

# U–Pb geochronology of epidote by LA–ICP–MS as a tool for dating hydrothermal-vein formation

Veronica Peverelli[1], Tanya Ewing[1], Daniela Rubatto[1], Martin Wille[1], Alfons Berger[1], Igor Maria Villa[1,2], Pierre Lanari[1], Thomas Pettke[1], Marco Herwegh[1]

[1]Institute of Geological Sciences, University of Bern, Bern, 3012, Switzerland
[2]Dipartimento di Scienze dell'Ambiente e della Terra, University of Milano-Bicocca, Milan, 20126, Italy

*Correspondence to*: Veronica Peverelli (veronica.peverelli@geo.unibe.ch)

**Abstract.** Monoclinic epidote is a low-μ (μ = $^{283}U/^{204}Pb$) mineral occurring in a variety of geological environments, participating in many metamorphic reactions and stable throughout a wide range of pressure–temperature conditions. Despite containing fair amounts of U, its use as a U–Pb geochronometer has been hindered by the commonly high contents of initial Pb with isotopic compositions that cannot be assumed a priori. We present U–Pb geochronology of hydrothermal-vein epidote spanning a wide range of Pb (3.9–190 μg g⁻¹), Th (0.009–38 μg g⁻¹) and U (2.6–530 μg g⁻¹) contents and with μ values between 7–510 from the Albula area (eastern Swiss Alps), from the Grimsel area (central Swiss Alps) and from the Heyuan fault (Guangdong province, China). The investigated epidote samples show appreciable fractions of initial Pb that vary to different extents. A protocol has been developed for in situ U–Pb dating of epidote by spot-analysis laser ablation inductively coupled plasma mass spectrometry (LA–ICP–MS) with a magmatic allanite as primary reference material. The suitability of the protocol and the reliability of the measured isotopic ratios have been ascertained by independent measurements of $^{238}U/^{206}Pb$ and $^{207}Pb/^{206}Pb$ ratios respectively by quadrupole and multicollector ICP–MS applied to epidote micro-separates digested and diluted in acids. For age calculation, we used the Tera–Wasserburg ($^{207}Pb/^{206}Pb$–$^{238}U/^{206}Pb$) diagram, which does not require corrections for initial Pb and provides the initial $^{207}Pb/^{206}Pb$ ratio if all intra-sample analyses are co-genetic. Petrographic and microstructural data indicate that the calculated ages date the crystallization of vein epidote from a hydrothermal fluid and that the U–Pb system was not reset to younger ages by later events. Vein epidote from the Albula area formed in the Paleocene (62.7 ± 3.0 Ma) and is related to Alpine greenschist-facies metamorphism. The Miocene (19.1 ± 4.0 Ma and 16.9 ± 3.7 Ma) epidote veins from the Grimsel area formed during the Handegg phase (22–17 Ma) of the Alpine evolution of the Aar Massif. Identical initial $^{207}Pb/^{206}Pb$ ratios reveal homogeneity in Pb isotopic compositions of the fluid across ca. 200 m. Vein epidote from the Heyuan fault is Cretaceous in age (108.1 ± 8.4 Ma) and formed during the early movements of the fault. In situ U–Pb analyses of epidote returned reliable ages of otherwise undatable epidote-quartz veins. The Tera–Wasserburg approach has proven pivotal for in situ U–Pb dating of epidote and the decisive aspect for low age uncertainties is the variability in intra-sample initial Pb fractions.



## 1 Introduction

Linking petrological and structural information to the timing of geological events is crucial to better constrain the sequence of geodynamic processes. In this context, the role of fluids in the continental crust is particularly relevant because they mediate and influence deformation and metamorphism (e.g. Wyllie, 1977; Etheridge et al., 1983; Johannes, 1984; Pennacchioni and Cesare, 1997; Malaspina et al., 2011; Wehrens et al., 2016). The formation of a hydrothermal vein

represents a specific deformation and hydration event in the geological history of the host rock, during which the vein-filling minerals record the geochemical signature of the mineralizing fluid (e.g. Elburg et al., 2002; Barker et al., 2009; Bons et al., 2012; Parrish et al., 2018; Ricchi et al., 2019; 2020). By combining different geochemical and geochronological techniques to suitable vein-filling minerals it is therefore possible to determine when the vein formed and the isotopic signature of the fluids for insight into their origin (e.g. Pettke et al., 2000; Barker et al., 2006; Elburg et al., 2002).

Monoclinic epidote [$Ca_2Al_3Si_3O_{12}(OH)$–$Ca_2Al_2Fe^{3+}Si_3O_{12}(OH)$] is a common rock-forming and vein-filling mineral (e.g. Bird and Spieler, 2004; Franz and Liebscher, 2004; Guo et al., 2014; Zanoni et al., 2016). It is stable over a wide range of pressure–temperature conditions and in a multitude of magmatic, metamorphic and hydrothermal mineral assemblages (Bird and Spieler, 2004; Enami et al., 2004; Grapes and Hoskin, 2004; Schmidt and Poli, 2004). Its complex crystal structure incorporates of a large variety of elements, enabling the measurements of trace element (e.g. Frei et al., 2004) and isotopic

(e.g. Guo et al., 2014) signatures. Uranium and thorium are readily accepted into the epidote structure, with contents that are highly variable but generally in trace element levels (Frei et al., 2004). Hence, attempts have been made at using it as a geochronometer by step-wise leaching Pb–Pb dating (e.g. Buick et al., 1999) and thermal ionization mass spectrometry (TIMS) U–Pb dating (e.g. Oberli et al., 2004). Buick et al. (1999) constrained the timing of vein formation and that of subsequent fluid pulses in garnet-epidote-quartz veins in the Reynolds Range (central Australia). Oberli et al. (2004)

obtained a U–Pb age of magmatic epidote from the Bergell pluton (eastern Central Alps) and identified epidote formation as a late-stage process during the solidification of the pluton. However, none of these techniques are in situ, and they cannot target different microstructural and textural domains. Monoclinic epidote could provide valuable geochronological and isotopic information where no other datable minerals are available. Good examples are epidote-quartz veins that are widespread in the Alps (e.g. Aar Massif and Albula area). Syn-kinematic epidote in breccias associated with rift-related

faults in the Campos basin (southeastern Brazil) may highlight successive phases of fault movement (Savastano et al., 2017). In the Zermatt-Saas Zone (Western Alps), epidote is a peak-pressure rock-forming mineral in epidote-bearing rodingites (Zanoni et al., 2016) and it may help to better constrain the P–T–d–t paths related to pressure-peak metamorphism.

This contribution discusses the applicability of in situ U–Pb dating to monoclinic epidote. To our knowledge, no analytical protocols have been proposed in this respect. To fill this gap, we present U–Pb ages measured in hydrothermal-vein epidote

by LA–ICP–MS using a magmatic allanite for standardization. The main issues related to the proposed geochronometer and addressed in this contribution are 1) the suitability of magmatic allanite as the most closely matrix-matched reference material for LA–ICP–MS U–Pb dating of monoclinic epidote in spot-analysis mode, 2) the applicability of the protocol with





respect to the different contents of initial Pb fractions and U in the studied samples and 3) the effects on age precision of the interplay between analyzed volumes and preservation of chemical variability. The Tera–Wasserburg diagram proves to be

the key tool for successful epidote U–Pb geochronology, allowing to add monoclinic epidote to the list of low-μ U–Pb geochronometers. Notably, by investigating epidote on its own, it is possible to combine U–Pb ages and isotopic systematics with data from trace element analyses and other isotopic systems; this may permit to reconstruct fluid flow and its origin with information that is all provided by a single mineral.

## 1.1 The challenges of investigating monoclinic epidote as a geochronometer

Along with relevant amounts of $U^{4+}$ and $Th^{4+}$ as Ca substitution in the A site and of $U^{6+}$ as Al or Fe substitution in the M site (Frei et al., 2004), high contents of initial Pb are incorporated by epidote during crystallization. This causes the dilution of ingrown radiogenic Pb, whose measurements are imperative for U–Pb geochronology, and makes epidote a low parent-to-daughter or low-μ phase (i.e. μ < ca. 2000; Romer, 2001; Romer and Xiao, 2005). U–Pb dating of initial Pb-rich minerals can proceed in two ways depending on whether or not the isotopic composition of the initial Pb is known or can be

reasonably assumed. Assumptions can be based on the modeled evolution of global Pb isotopic compositions such as those proposed by Cumming and Richards (1975) and Stacey and Kramers (1975). In the first case, a correction for initial Pb can be applied, and an initial Pb-corrected U–Pb age can be calculated from the measured U(±Th)–Pb isotopic ratios of each analysis (Williams, 1998). However, age inaccuracies due to wrong assumptions regarding initial Pb isotopic compositions are remarkable (see Romer, 2001; Romer and Xiao, 2005). An initial Pb-correction can be applied without any assumptions

if the contents of $^{204}Pb$ – the only non-radiogenic lead isotope – can be measured precisely. If the isotopic composition of initial Pb is unknown, $^{204}Pb$ contents cannot be precisely determined (e.g. because of the analytical technique employed) and no other dating method is viable (e.g. too-low Th contents hampering Th–Pb dating), a valuable tool for U–Pb dating of low-μ phases is the Tera–Wasserburg diagram (Tera and Wasserburg, 1972) that plots measured $^{207}Pb/^{206}Pb$ versus $^{238}U/^{206}Pb$ ratios. Its advantages are that 1) it does not require corrections for initial Pb isotopic compositions, 2) it provides the initial

$^{207}Pb/^{206}Pb$ ratio itself in addition to a $^{207}Pb$-corrected U–Pb age, and 3) it gives an estimate of the fractions of initial lead relative to those of radiogenic Pb in each analysis (Tera and Wasserburg, 1972; Ludwig, 1998). This approach is based on the hypotheses that multiple analyses are performed on material of the same age and sharing the same initial Pb isotopic composition. If these criteria are met, one regression is defined by the alignment of the measurements of $^{207}Pb/^{206}Pb$ vs. $^{238}U/^{206}Pb$ ratios, whose lower intercept with the concordia yields the age of the sample. If the hypotheses prove to be wrong

(e.g. multiple mineral generations), this is highlighted by the statistical parameters of the regression. The fraction of initial Pb in each analysis can be estimated from the proximity of individual data points to the upper $^{207}Pb/^{206}Pb$ intercept of the regression (Tera and Wasserburg, 1972; Ludwig, 1998) which gives the initial $^{207}Pb/^{206}Pb$ ratio of the sample. The Tera-Wasserburg approach, however, is not suitable for all initial Pb-rich mineral. In fact, key to a successful use of this diagram is a reasonable spread in plotted data points, which depends on the variability in initial Pb fractions and U contents. If the

analyzed mineral has poorly variable fractions of initial Pb and U concentrations, the regression cannot be well constrained



and age precision is low. Epidote minerals are commonly characterized by chemical zoning (Franz and Liebscher, 2004), which may also reflect variability in initial Pb fractions and U contents and may promote spread of the data points along the Tera–Wasserburg regression.

A suitable technique for in situ U–Th–Pb dating is LA–ICP–MS in spot-analysis mode, provided that U/Pb and Th/Pb
elemental fractionation at the ablation site (downhole fractionation, DF) is appropriately corrected for over ablation time by relying on an external reference material (e.g. Sylvester, 2005; Košler, 2007; McFarlane et al., 2016). Since DF is matrix dependent (e.g. Sylvester, 2005; Košler, 2007; Sylvester, 2008; El Korh, 2014), a matrix-matched reference material is most commonly used. To date, no reference epidote exists, posing the problem of correction for DF of $^{238}$U/$^{206}$Pb ratios measured in epidote, crucial for accurate age determinations by LA–ICP–MS (Horstwood et al., 2016). However, in recent years,
magmatic allanite has been successfully characterized and dated by U–Th–Pb LA–ICP–MS (e.g. Gregory et al., 2007; 2012; El Korh, 2014; Smye et al., 2014), and several allanite samples have been proposed as suitable primary reference materials (e.g. Gregory et al., 2007; Smye et al., 2014). Allanite [(Ca, REE, Th)$_2$(Fe$^{3+}$, Al)$_3$Si$_3$O$_{12}$(OH)] is the REE-rich member of the epidote mineral group with ThO$_2$ contents of 2–3 wt% and U concentrations often below 1000 ppm (Gieré and Sorensen, 2004; and references therein), and it is a promising candidate as a closely matrix-matched reference material for monoclinic
epidote. The possible issues in the use of allanite as reference material for accurate U–Th–Pb geochronology are mostly related to local isotopic heterogeneity, excess $^{206}$Pb due to incorporation of $^{230}$Th during crystallization, variable contents of initial Pb and disturbance of the geochronometer by secondary processes (e.g. hydrothermal alteration; Gregory et al., 2007; Darling et al., 2012; Smye et al., 2014; Burn et al., 2017). Nevertheless, these issues can be largely avoided by careful selection of spot analyses referring to backscattered electron (BSE) images, and by identifying and excluding problematic
analyses from calculations. A disadvantage of LA–ICP–MS is the large isobaric interference on mass 204 by $^{202}$Hg of the carrier gas. A correction for such an interference in order to apply a $^{204}$Pb-correction is complex, and therefore leaves one with the only option of the application of the Tera–Wasserburg approach. In this study, epidote ages and initial $^{207}$Pb/$^{206}$Pb ratios are assessed from the Tera–Wasserburg diagram. If initial $^{207}$Pb/$^{206}$Pb ratios are consistent with modeled values of initial Pb isotopic compositions (e.g. Stacey and Kramers, 1975), then an accurate $^{238}$U/$^{206}$Pb age can be obtained by
averaging single-spot ages, which are calculated from each analysis corrected for initial Pb by applying a $^{207}$Pb-correction (i.e. weighted average $^{207}$Pb-corrected $^{238}$U/$^{206}$Pb age; see Williams, 1998).





## 2 Geological context and field relations

Figure 1: Scans of thin sections of (a) Albula-1, (b) Grimsel-1, (c) Grimsel-2 and (d) Heyuan-1 samples. (a) and (b): plane polarized
light on petrographic microscope; (c) and (d): plane light. bt = biotite; chl = chlorite; ep = epidote; kfs = K-feldspar; plg =
plagioclase; qtz = quartz.

Hydrothermal epidote veins (Fig. 1) were sampled at Albula Pass (eastern Swiss Alps), at Grimsel Pass (central Swiss Alps)
and at the Heyuan Fault (Guangdong Province, China). Sampling locations are respectively shown in Fig. 1 of Şengör
(2016), in Fig. 1 of Wehrens et al. (2017), and in Figs. 1 and 2 of Tannock et al. (2020a). The Albula area was chosen
because the weak Alpine overprint allows for the hypothesis that epidote veins were not geochemically and isotopically
altered after their formation. The Grimsel area is advantageous because of the vast knowledge about the tectono-
metamorphic events that affected the region. The Heyuan Fault was selected because structural constraints allow to assess if
the calculated U–Pb age is reasonable for epidote crystallization.

The Albula area is located in the upper Err nappe, close to the tectonic contact with the Ela nappe. It belongs to the
Austroalpine domain, the basement of the former Adriatic continental margin (e.g. Froitzheim and Eberli, 1990; Froitzheim
et al., 1994). The most common lithology in the Err basement is the Albula Granite, a granodiorite of Variscan to post-



Variscan age (e.g. Manatschal and Nievergelt, 1997; Incerpi et al., 2017) and in which epidote ± quartz veins are widespread. In the late Carboniferous and early Permian, the Albula Granite intruded into the metamorphic basement of the Err nappe at < 3 km depth (Mohn et al., 2011; and references therein). Subsequently, the Lower Austroalpine was involved in the Jurassic
rifting that led to the break-up of Pangea (e.g. Manatschal et al., 2000). During the Alpine orogeny, the Err nappe mainly recorded the deformation resulting from the W- to NW-directed vergence of the Austroalpine domain from Cretaceous until early Cenozoic times with maximum metamorphic conditions reaching lower-greenschist facies, and was only weakly affected by the Cenozoic tectonics (e.g. Froitzheim and Manatschal, 1996; Mohn et al., 2011; Epin et al., 2017).

The Grimsel area is in the Aar Massif, one of the External Crystalline Massifs of the Alps (Rolland et al., 2009; Wehrens et
al., 2017; Herwegh et al., 2020). Here, epidote-quartz veins are common in the Central Aar granite and in the Grimsel granodiorite, which during the earliest Permian intruded into a polycyclic basement bearing evidence of Ordovician metamorphism and Variscan overprint (Schaltegger and Corfu, 1992; Berger et al., 2017). After being affected by the Jurassic rifting, the Aar Massif was involved in the continent–continent collision during the Alpine orogeny, as demonstrated by the presence of anastomosing high-strain shear zones of Alpine age (e.g. Goncalves et al., 2012; Wehrens et al., 2017).
Metamorphism at greenschist facies conditions reached 450 ± 30 °C at 0.6 ± 0.1 kbar in this area (Challandes et al., 2008; Goncalves et al., 2012). The Alpine history of the Aar Massif is subdivided into three phases: 1) the Handegg phase (22–17 Ma; Challandes et al., 2008), with stable green biotite in the shear zones (Challandes et al., 2008; Rolland et al., 2009; Herwegh et al., 2017; Wehrens et al., 2017); 2) the Oberaar phase in the southern Aar Massif (14–3.4 Ma; Hofmann et al., 2004; Rolland et al., 2009), with mica and chlorite stable in the shear zones, and metastable biotite (Herwegh et al., 2017;
Wehrens et al., 2017); 3) the Pfaffenchopf phase in the northern Aar Massif (< 12 Ma; Herwegh et al., 2020). The epidote-quartz veins analyzed in this study were sampled at a distance of ca. 200 m in the Nagra Felslabor tunnel at Grimsel Pass. As these veins are only visible within the tunnel, their relationships with Alpine structures and between each other are not understood.

The Heyuan Fault is a crustal-scale fault that formed in Mesozoic times as a low-angle normal fault, but is currently active
under a transpressive regime (Tannock et al., 2020a; 2020b). The footwall of this fault mainly consists of the Xinfengjiang pluton (the eastern portion of the Fogang batholith), a late Jurassic biotite granite that intruded into the basement of Proterozoic to Silurian age during the Yanshanian orogeny (Li et al., 2007; Tannock et al., 2020a; 2020b). Epidote veins are located in the mylonites at the transition between undeformed granite and fault zone (Tannock et al., 2020a). The hanging wall is composed of a quartz-sericite ultracataclasite/phyllonite in contact with a quartz reef and finally abutted by the
sedimentary "Red Beds" of Cretaceous age (Tannock et al., 2020a). Since the epidote veins are either pre- or syn-kinematic with respect to the mylonites (Tannock et al., 2020b), we infer that the epidote veins cannot be older than the pluton itself, but they are also among the earliest structures related to the early movements of the Heyuan Fault (Tannock et al., 2020a; 2020b). Epidote veins are absent in the footwall cataclasite and in the quartz reef, which formed after the mylonite. Syn-kinematic epidote veins formed at a temperature of ca. 330 °C as indicated by the white mica composition in the mylonites
(Tannock et al., 2020a).





## 3 Methods

Except where stated, sample preparation and measurements were carried out in the laboratory facilities of the Institute of Geological Sciences, University of Bern, Switzerland.

### 3.1 Imaging and screening methods for sample selection

Thin (30 μm) and thick (50–60 μm) sections were inspected by petrographic and electron microscopy, respectively on a ZEISS Axioplan microscope and on a ZEISS EVO50 SEM using BSE imaging (ca. 1 nA beam current, 20 kV accelerating voltage). BSE images were used to plan analysis spots – all of the same size – within epidote grains so as to avoid mixing of different zonings in each single measurement, as well as mineral and fluid inclusions. U, Th and Pb contents were measured by LA–ICP–MS upon screening many samples, employing methods presented elsewhere (Pettke et al., 2012). The details of
the LA–ICP–MS setup are reported in Appendix A.

### 3.2 U–Pb geochronology by LA–ICP–MS

Isotopic measurements of U, Th and Pb were performed on thin/thick sections for epidote and on acryl grain mounts for allanite. To minimize surface contamination, the thin/thick sections were cleaned with ethanol, and the grain mounts with ethanol and 5 % $HNO_3$. Measurements of U, Th and Pb isotopic ratios were performed with a Resonetics RESOlutionSE 193
nm excimer laser system (Applied Spectra, USA) equipped with a S-155 large-volume constant-geometry chamber (Laurin Technic, Australia) coupled with an Agilent 7900 ICP–QMS. The suitability of analytical conditions (Table 1) was checked by performing preliminary analyses on secondary reference materials of known ages (CAP[b] allanite; Burn et al., 2017; CAP and AVC allanites; Barth et al., 1994) and comparing them to their reference values. Low fluence of 3 J s$^{-2}$, low repetition rate of 5 Hz and large spot size of 50 μm were combined to ensure a slow increase in depth/diameter ratio of the laser crater
during a 40 sec ablation time, hence to minimize elemental (U/Th/Pb) DF.

Table 1: Measurement conditions on Agilent 7900 for U–Th–Pb isotopic data by LA–ICP–MS.

| Parameters | 14 June 2019 | 23 July 2019 | 16 January 2020 |
|---|---|---|---|
| RF power | 1280 W | 1320 W | 1380 W |
| Fluence | | 3 J cm$^{-2}$ | |
| Repetition rate | | 5 Hz | |
| Cell gas flow | 3.0 ml min$^{-1}$ $N_2$ and 350 ml min$^{-1}$ He | | 3.0 ml min$^{-1}$ $N_2$ and 400 ml min$^{-1}$ He |
| Sensitivity on mass 232 (beam size, fluence, repetition rate) | ca. 1'500'000 cps (50 μm, 2.5 J cm$^{-2}$, 5 Hz) | ca. 1'600'500 cps (50 μm, 2.5 J cm$^{-2}$, 5 Hz) | ca. 1'300'400 cps (50 μm, 2.5 J cm$^{-2}$, 5 Hz) |
| 232/238 ratio | | > 0.97 | |



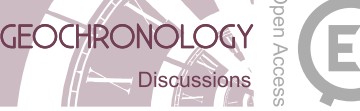

| 248/232 ratio | < 0.002 | |
|---|---|---|
| Background | 30 sec | |
| Pre-cleaning (beam size in µm) | 10 pulses (64); followed by wait time of 10 s before ablation | 10 pulses (30); followed by wait time of 10 s before ablation |
| Ablation time (beam size in µm) | 40 sec (50) | 40 sec (30) |
| Measured masses (dwell times in ms) | 204 (40), 206 (40), 207 (40), 208 (40), 232 (40), 238 (40) | 206 (40), 207 (40), 208 (40), 232 (40), 238 (40) |
| Primary standards | Tara allanite | |
| Secondary standards | CAP[b] allanite | CAP and AVC allanites |

An additional session was carried out with a laser spot of 30 µm and the same laser conditions. The aim was to assess the
effects of using a smaller spot size on the correction for DF (see Chew et al., 2014) and to explore whether the use of allanite
as primary reference material can still provide accurate data. If so, this would extend the applicability of the present protocol
to smaller epidote grains. For this test, we selected two samples: the one also used for solution ICP–MS measurements (see
Sect. 3.3), and the one with the smallest averaged analytical errors on $^{238}$U/$^{206}$Pb and $^{207}$Pb/$^{206}$Pb ratios. The 30 µm analyses
were done in the same areas and crystals analyzed in the previous sessions after polishing the thin/thick sections to remove
the condensation blankets around the ablation craters. In all sessions, Tara allanite (see Gregory et al., 2007; Smye et al.,
2014) is chosen as primary reference material because it is the most homogenous allanite in terms of U–Th–Pb isotopes
(Gregory et al., 2007; Burn et al., 2017) and the most promising reference material for U–Pb geochronology (Smye et al.,
2014). Tara allanite reference isotopic ratios and their uncertainties (Table 2) were calculated by averaging five ID–TIMS
measurements reported by Smye et al. (2014), excluding the measurement that yielded the youngest U–Pb age outside
uncertainty (Smye et al., 2014). The analytical sequence involved measurements of the reference Tara allanite interspersed
between 3–9 sample measurements including allanite secondary reference materials for quality control. Analysis spots in
allanite were planned based on BSE images to avoid chemical and isotopic heterogeneity (i.e. mixing of zoning) within each
single analysis, and inclusions (e.g. rare < 1 µm sized thorite; Smye et al., 2014).

**Table 2: Reference data of Tara allanite for normalization of U–Th–Pb isotopic data by LA–ICP–MS. The ratios are averages
calculated from the measurements by Smye et al. (2014) by ID–TIMS; one measurement was excluded (see text). Uncertainties are
given in brackets and are calculated as 2 standard errors. Subscript c = common.**

| Isotopic ratio | Tara allanite |
|---|---|
| $^{206}$Pb/$^{238}$U | 0.0678 (0.0008) |
| $^{207}$Pb/$^{235}$U | 0.5020 (0.0173) |
| $^{208}$Pb/$^{232}$Th | 0.0195 (0.0021) |
| $^{207}$Pb/$^{206}$Pb | 0.0537 (0.0016) |





| $(^{207}Pb/^{206}Pb)_c$ | 0.866 (0.079) |
|---|---|

Raw data were treated in the software Iolite (version 7.08) by the VisualAge_UcomPbine Data Reduction Scheme (Chew et
al., 2014). The quality of signals and that of the correction for DF were considered to determine the validity of each
measurement. Assessing the quality of signals implies inspection of the laser signal of each isotope across each measurement
to discard – partially or entirely – those that are contaminated by impurities such as mineral or fluid inclusions, or that show
isotopic heterogeneity during ablation. Assessing the correction for DF is based on the fact that if the ablation behavior is the
same between primary reference material and sample, the correction for DF of the isotopic ratios measured in the latter is
accurately carried out, as assessed by their DF-corrected $^{206}Pb/^{238}U$ ratios being flat during ablation. This and the subsequent
normalization of the measured ratios based on the reference values of the primary reference material ensure that the
$^{238}U/^{206}Pb$ and $^{207}Pb/^{206}Pb$ ratios used in the Tera–Wasserburg diagrams are true values, and that the U–Pb age calculated
with these ratios is accurate. Since the isotopic fractionation between $^{207}Pb$ and $^{206}Pb$ is negligible (e.g. Burn et al., 2017), we
address the suitability of the DF correction based on Tara allanite as primary reference material only on $^{206}Pb/^{238}U$ ratios. A
$^{207}Pb$-correction was applied to the primary reference material (i.e. Tara allanite) using an initial $^{207}Pb/^{206}Pb$ value of 0.864 ±
0.015 (420 Ma) following Stacey and Kramers (1975). The need to correct measurements of Tara allanite as primary
reference material for initial Pb using such a value was proposed by Gregory et al. (2007) and confirmed by Smye et al.
(2014). Isoplot 3.7.5 (Ludwig, 2012) was used for age calculations. Age determination of epidote samples and allanite
secondary reference materials relies on the Tera–Wasserburg approach (Tera and Wasserburg, 1972; Ludwig, 1998). Since
the initial Pb isotopic composition of CAP, CAP[b] and AVC allanite is known and consistent with a modeled two-stage
evolution of initial Pb isotopic compositions (Barth et al., 1994; Gregory et al., 2007; Burn et al., 2017), we ensured better
age precision by anchoring their Tera–Wasserburg regressions to an initial $^{207}Pb/^{206}Pb$ ratio of 0.854 ± 0.015 (275 Ma;
Stacey & Kramers, 1975) and calculated their weighted average $^{207}Pb$-corrected $^{238}U/^{206}Pb$ ages using the same value.
Regression and weighted average $^{207}Pb$-corrected $^{238}U/^{206}Pb$ ages of allanite secondary reference materials are summarized in
Table 3, and their Tera–Wasserburg diagrams are presented in Fig. B1 (Appendix B). Allanite secondary standard age
uncertainties are 95 % confidence.

**Table 3: U–Pb LA–ICP–MS ages of allanite secondary standards measured in three analytical sessions in this study. Age uncertainties are 95 % confidence.**

| Sample | 14 June 2019 | | 23 July 2019 | | 16 January 2020 | |
|---|---|---|---|---|---|---|
| | Isochron U–Pb age [Ma] | Weighted average U–Pb age [Ma] | Isochron U–Pb age [Ma] | Weighted average U–Pb age [Ma] | Isochron U–Pb age [Ma] | Weighted average U–Pb age [Ma] |
| CAP[b] | 284.2 ± 2.6 MSWD = 0.34 | 284.2 ± 2.0 MSWD = 0.34 | - | - | - | - |



| | | | | | | |
|---|---|---|---|---|---|---|
| CAP | - | - | 288.5 ± 2.9<br>MSWD = 1.04 | 288.6 ± 2.3<br>MSWD = 1.03 | 283.0 ± 3.4<br>MSWD = 1.2 | 282.5 ± 3.2<br>MSWD = 1.2 |
| AVC | - | - | 292.4 ± 3.7<br>MSWD = 0.49 | 292.2 ± 2.3<br>MSWD = 0.48 | 285.2 ± 4.5<br>MSWD = 0.70 | 285.1 ± 3.5<br>MSWD = 0.69 |

## 3.3 Solution ICP–MS

Independent measurements of $^{238}U/^{206}Pb$ and $^{207}Pb/^{206}Pb$ ratios were performed on two epidote micro-separates to check their consistency with U–Pb isotopic data measured by LA–ICP–MS and hence the reliability of the latter data. The material was separated from the epidote-quartz vein of the sample from the Albula area, which is the one with the lowest degree of deformation and largest epidote crystals (see Sect. 4.1). Clean and pure epidote grains were handpicked under a binocular microscope. The epidote separates were pre-cleaned with MilliQ™ water. Based on LA–ICP–MS U and Pb concentration data, four sample aliquots – two from each epidote micro-separate and each corresponding to ca. 300 ng of total Pb – were weighed in acid-cleaned Teflon beakers and dissolved following the procedure of Nägler and Kamber (1996). Samples were leached with aqua regia at 120 °C for two days. The leachate was transferred into a second pre-cleaned Teflon beaker. To ensure complete dissolution a concentrated $HF:HNO_3$ (3:1 by volume) was added to the supernatant, and the beakers were placed on a hot plate at 90 °C for two days. After drying, 2 ml of 6.4 M HCl were added, and the beakers were placed on a heating plate at 150 °C for two days. The same procedure was applied to standard AGV-2 (Weis et al., 2006) as well as to two blanks, and complete dissolution was achieved for all samples and standards. Finally, the samples were dissolved in 1 ml 0.5 M $HNO_3$.

To determine $^{238}U/^{206}Pb$ ratios, a 10 % aliquot of digested samples and standards was further diluted with 0.5 M $HNO_3$ up to a final volume of 10 ml. Two solutions with two different dilution factors were prepared from each sample aliquot and were analyzed on a 7700x Agilent™ quadrupole ICP–MS at the Department of Geography, University of Bern, Switzerland. Standard AGV-2 (Weis et al., 2006) was used to correct for instrumental fractionation and to check accuracy of measurements. Final sample concentrations of $^{206}Pb$, $^{207}Pb$ and $^{238}U$ – for both dilution factors of each sample aliquot – and their corresponding analytical uncertainties as relative standard deviations – solely based on counting statistics – were calculated by referring to a calibration curve based on three dilution factors of AGV-2 standard. The $^{238}U/^{206}Pb$ ratio and uncertainty as 2 SE of each sample aliquot were calculated with Isoplot 3.7.5 (Ludwig, 2012) as weighted average values between the $^{238}U/^{206}Pb$ ratios calculated from the measurements of both dilution factors, which were the same within uncertainty for all sample aliquots. The remaining sample material was dried and re-dissolved in 0.5 ml 1 M $HNO_3$ for Sr–Pb column chemistry using a pre-cleaned Sr-spec™ resin (Horwitz et al., 1992). After loading, the sample matrix was eluted from the column with 1.5 ml 1 M $HNO_3$ while Sr and Pb were retained on the column. The Sr and Pb fractions were eluted with 1 ml of 0.01 M $HNO_3$ and 8 ml of 0.01 M HCl respectively, following Villa (2009) and Quistini et al. (2017). After drying, the Pb fraction was dissolved and further diluted in 0.5 M $HNO_3$ for measurement of Pb isotopes on a Thermo Fisher Neptune Plus MC–ICP–MS. Measurements were carried out in dry plasma mode using a CETAC Aridus 2 desolvating



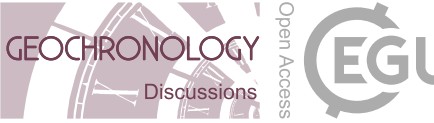

system. Thallium was added to samples and standards to correct for instrumental mass fractionation with repeated
measurements of NIST SRM 981 to quantify the external reproducibility of the measurements (Villa, 2009); the measured
Pb isotopic composition was indistinguishable from those reported by Rehkämper and Mezger (2000). The four pairs of
isotopic ratios measured by solution ICP–MS are only compared to the Tera–Wasserburg diagram based on LA–ICP–MS
data (50 µm spot size) and are not used to calculate an age because for an unanchored regression to be statistically robust a
minimum of 15 analyses is necessary.

**4 Results**

**4.1 Petrography and U–Th–Pb contents of samples selected for U–Pb geochronology**

Four samples were selected for this contribution mainly based on 1) size of epidote grains in order to use the largest laser
beam possible for LA–ICP–MS and 2) U contents that are both as high as possible and as variable as possible within the
sample. Larger laser beams maximize the precision of U–Pb geochronology measurements. High U contents are related to
higher contents of uranogenic Pb isotopes and therefore improve the precision of U and Pb isotopic measurements; their
variability contributes to a larger spread of the analyses in Tera–Wasserburg diagrams for well-constrained regressions.

One sample from the Albula area, sample Albula-1, was selected for U–Pb geochronology. Two veins can be recognized
(Fig. 1a), both crosscutting the host rock with sharp boundaries:

1) 2–3 cm wide epidote-quartz-plagioclase vein (Vein1). Epidote grains are elongated, with lengths between ca. 0.5 mm
along the vein boundaries to ca. 1 cm towards the center of the vein, with an aspect ratio up to ca. 7:1. Fractures are common
and grains are euhedral to subhedral. Quartz is fractured and plagioclase is limited to a ca. 2 mm wide portion along the vein
boundaries, associated with the smallest epidote grains. U contents of epidote range between 3.7–89 µg g$^{-1}$ (Table 4). Th
contents are 0.009–0.047 µg g$^{-1}$ (19/25 measurements are below the limits of detection of 0.03–0.07 µg g$^{-1}$ with a spot size of
24 µm and 0.003 µg g$^{-1}$ with a 60 µm spot size). Pb contents are 3.9–62 µg g$^{-1}$, total Pb/U ratios 0.14–10 and µ values 7–510.


**Table 4: Concentrations of Pb, Th and U, Th/U and Pb/U ratios, and µ values measured by laser ablation ICP–MS with the trace
element protocol in Appendix A. The symbol < is followed by limits of detection (calculated for each element in each measurement
individually following the formulation in Pettke et al., 2012) for those analyses below it with spot size indicated in brackets (in µm).
µ values are calculated from total Pb and total U contents by considering an isotopic abundance of 1.4 % for $^{204}$Pb and 93 % for**
**$^{238}$U. N/A = non-applicable.**

| Sample | Pb | Th | U | Th/U | Pb/U | µ = $^{238}$U/$^{204}$Pb |
|---|---|---|---|---|---|---|
|  | µg g$^{-1}$ | µg g$^{-1}$ | µg g$^{-1}$ | - | - | - |
| **Albula-1** | 17 | 0.02 | 89 | 0.0002 | 0.20 | 360 |
| **Vein1** | 19 | 0.04 | 22 | 0.002 | 0.87 | 81 |
|  | 6.5 | <0.025 | 11 | N/A | 0.60 | 120 |
|  | 7.9 | 0.05 | 12 | 0.004 | 0.66 | 110 |
|  | 7.9 | 0.03 | 8.9 | 0.004 | 0.89 | 80 |





| | | | | | |
|---|---|---|---|---|---|
| | 3.9 | <0.014 | 28 | N/A | 0.14 | 510 |
| | 9.7 | <0.005 | 28 | N/A | 0.35 | 210 |
| | 18 | <0.022 | 7.1 | N/A | 2.5 | 28 |
| | 21 | 0.03 | 11 | 0.003 | 1.8 | 39 |
| | 11 | <0.004 | 16 | N/A | 0.70 | 100 |
| | 8.2 | <0.072 | 11 | N/A | 0.74 | 96 |
| | 8.5 | <0.030 | 3.7 | N/A | 2.3 | 31 |
| | 7.5 | <0.048 | 4.5 | N/A | 1.7 | 43 |
| | 10 | <0.049 | 5.9 | N/A | 1.7 | 42 |
| | 16 | <0.039 | 5.5 | N/A | 2.8 | 25 |
| | 24 | <0.064 | 8.3 | N/A | 2.9 | 25 |
| | 5.7 | <0.030 | 23 | N/A | 0.25 | 290 |
| | 61 | <0.007 | 6.0 | N/A | 10 | 6.9 |
| | 46 | <0.0054 | 7.5 | N/A | 6.1 | 12 |
| | 9.0 | <0.018 | 3.7 | N/A | 2.4 | 29 |
| | 6.4 | <0.064 | 12 | N/A | 0.51 | 140 |
| | 8.3 | 0.01 | 26 | 0.0003 | 0.32 | 220 |
| | 7.6 | <0.007 | 27 | N/A | 0.28 | 260 |
| | 6.9 | <0.008 | 9.6 | N/A | 0.72 | 99 |
| | 25 | <0.003 | 4.3 | N/A | 5.9 | 12 |
| **Albula-1** | 44 | 0.67 | 26 | 0.03 | 1.7 | 43 |
| **Vein2** | 64 | 14 | 140 | 0.10 | 0.46 | 160 |
| | 24 | 4.0 | 36 | 0.11 | 0.69 | 100 |
| **Grimsel-1** | 130 | 0.12 | 190 | 0.001 | 0.68 | 100 |
| | 150 | 7.6 | 180 | 0.04 | 0.83 | 85 |
| | 170 | 0.15 | 270 | 0.001 | 0.64 | 110 |
| | 150 | 0.99 | 230 | 0.004 | 0.67 | 110 |
| | 130 | 0.47 | 79 | 0.01 | 1.6 | 44 |
| | 130 | <0.094 | 130 | N/A | 1.0 | 70 |
| | 150 | 0.82 | 350 | 0.002 | 0.44 | 160 |
| | 97 | 0.07 | 60 | 0.001 | 1.6 | 44 |
| | 170 | 0.71 | 130 | 0.01 | 1.3 | 52 |
| | 87 | 0.30 | 140 | 0.002 | 0.64 | 110 |
| | 170 | 0.56 | 240 | 0.002 | 0.72 | 99 |
| | 150 | 2.6 | 100 | 0.03 | 1.4 | 49 |
| | 170 | 0.47 | 220 | 0.002 | 0.78 | 91 |
| | 93 | 4.9 | 100 | 0.05 | 0.90 | 79 |
| | 190 | 0.81 | 140 | 0.01 | 1.4 | 50 |
| | 78 | 1.9 | 160 | 0.01 | 0.49 | 140 |
| | 180 | 0.14 | 260 | 0.001 | 0.68 | 104 |



|  |  |  |  |  |  |  |
|---|---|---|---|---|---|---|
|  | 93 | 0.04 | 54 | 0.001 | 1.7 | 41 |
|  | 150 | 0.84 | 140 | 0.01 | 1.1 | 65 |
| **Grimsel-2** | 75 | 0.07 | 220 | 0.0003 | 0.34 | 210 |
|  | 68 | 0.27 | 160 | 0.002 | 0.43 | 170 |
|  | 75 | 0.20 | 150 | 0.001 | 0.49 | 140 |
|  | 51 | <0.061 | 120 | N/A | 0.42 | 170 |
|  | 62 | <0.048 | 120 | N/A | 0.53 | 130 |
|  | 74 | 0.07 | 160 | 0.0004 | 0.46 | 150 |
|  | 97 | 0.11 | 270 | 0.0004 | 0.35 | 200 |
|  | 95 | 0.13 | 270 | 0.0005 | 0.36 | 200 |
|  | 65 | 0.21 | 280 | 0.001 | 0.23 | 310 |
|  | 88 | <0.022 | 180 | N/A | 0.50 | 140 |
|  | 93 | <0.034 | 110 | N/A | 0.86 | 83 |
|  | 75 | 0.53 | 535 | 0.001 | 0.14 | 510 |
|  | 62 | 0.36 | 310 | 0.001 | 0.20 | 350 |
| **Heyuan-1** | 19 | 0.48 | 8.4 | 0.06 | 2.2 | 32 |
|  | 12 | 0.30 | 18 | 0.02 | 0.68 | 100 |
|  | 16 | 0.38 | 8.7 | 0.04 | 1.8 | 39 |
|  | 15 | 0.42 | 8.4 | 0.05 | 1.8 | 39 |
|  | 27 | 0.67 | 7.5 | 0.09 | 3.6 | 20 |
|  | 16 | 0.05 | 13 | 0.00 | 1.2 | 57 |
|  | 21 | 17 | 24 | 0.70 | 0.86 | 82 |
|  | 20 | 38 | 34 | 1.10 | 0.58 | 120 |
|  | 26 | 0.08 | 18 | 0.00 | 1.4 | 50 |
|  | 17 | 0.15 | 7.0 | 0.02 | 2.4 | 29 |
|  | 16 | 1.4 | 9.7 | 0.14 | 1.7 | 42 |
|  | 19 | 0.13 | 13 | 0.01 | 1.5 | 49 |
|  | 23 | 18 | 30 | 0.62 | 0.78 | 90 |
|  | 17 | 0.12 | 9.1 | 0.01 | 1.8 | 39 |
|  | 17 | 1.3 | 7.0 | 0.19 | 2.4 | 29 |
|  | 24 | 0.16 | 14 | 0.01 | 1.6 | 43 |
|  | 22 | 2.2 | 9.9 | 0.22 | 2.2 | 32 |
|  | 8.2 | 0.04 | 22 | 0.00 | 0.37 | 190 |
|  | 18 | 0.88 | 8.3 | 0.11 | 2.2 | 31 |
|  | 12 | 0.22 | 2.6 | 0.08 | 4.4 | 16 |

2) ca. 1 mm wide epidote-quartz-plagioclase vein (Vein2). Epidote grains range between a few μm to 2 mm in diameter, most being fractured and euhedral to subhedral. Epidote grains of ca. 1–2 mm in diameter are wrapped by thin layers of μm-sized anhedral epidote grains. Quartz subgrains resulting from recrystallization and plagioclase wrap the epidote grains. U




contents of epidote are 26–140 µg g⁻¹ (Table 4), and Th contents 0.67–14 µg g⁻¹. Pb contents range from 24–64 µg g⁻¹, Pb/U ratios from 0.46–1.7 and µ values from 43–160.

BSE images of epidote (Fig. 2a) reveal growth zoning and intra-grain veinlets resulting from interaction with a secondary fluid. Sample Albula-1 was selected for solution ICP–MS given the large size of epidote grains.

**Figure 2: BSE images of (a) Albula-1, (b) Grimsel-1, (c) Grimsel-2 and (d) Heyuan-1 epidote. bt = biotite; chl = chlorite; ep = epidote; kfs = K-feldspar; plg = plagioclase; qtz = quartz.**

Sample Grimsel-1 (Fig. 1b) displays a folded epidote-quartz vein crosscutting a weakly deformed portion of the host rock. Epidote grains are generally prismatic and range between a few µm to ca. 2 mm in size. They are mostly subhedral to anhedral and cracked, and they form clusters with no preferential grain orientation. Quartz subgrains indicate dynamic recrystallization via subgrain rotation. Green biotite and rare chlorite are associated with the epidote-bearing vein. Epidote in BSE images (Fig. 2b) exhibits weak patchy zonations towards the rims and the presence of porosity. K-feldspar is





recognized within epidote cracks. U contents are 54–350 µg g$^{-1}$ (Table 4), and Th contents 0.041–4.9 µg g$^{-1}$. Pb contents range between 79–190 µg g$^{-1}$, with Pb/U ratios from 0.45–1.7 and µ values between 41–160.

Sample Grimsel-2 (Fig. 1c) consists of an epidote-quartz-biotite vein cutting through a weakly deformed sector of the host

rock. The vein boundaries are sharp and non-linear. Euhedral to subhedral epidote grains are cracked by stretching-induced fracturing, with single fragments ranging from a few µm to ca. 3 mm in size. Epidote grains can be estimated to have had an aspect ratio up to ca. 6:1 before fracturing. Quartz is recrystallized by subgrain rotation. Biotite grain sizes range between ca. 100–500 µm. BSE images (Fig. 2c) show that epidote exhibits regular growth zoning. U contents are 109–535 µg g$^{-1}$ and Th contents 0.066–0.53 µg g$^{-1}$ (Table 4). Pb contents range from 51–97 µg g$^{-1}$, Pb/U ratios are 0.20–0.86 and µ values 83–510.

Sample Heyuan-1 (Fig. 1d) is characterized by an epidote-quartz-K-feldspar-chlorite assemblage that fills pockets whose boundaries cut through the granite-forming minerals or follow their grain boundaries. The epidote-bearing assemblage is crosscut by quartz ± hematite veins (see Fig. 4c in Tannock et al., 2020b). Epidote is variably shaped, from elongated without preferential orientation to prismatic. Epidote ranges between tens of µm to ca. 2 mm in length and forms clusters of euhedral to anhedral crystals. Quartz associated with epidote is mostly recrystallized, as indicated by the presence of quartz

subgrains. Some mm-sized quartz grains, however, display undulose extinction. Chlorite associated with epidote forms interstitial aggregates of ca. 500–1000 µm in size. Growth zoning of epidote is recognized from BSE images (Fig. 2d) and K-feldspar is intertwined with smaller-sized epidote grains along the boundaries of larger ones, as well as with quartz filling epidote fractures. U contents are 2.6–34 µg g$^{-1}$ (Table 4). Th contents range between 0.042–38 µg g$^{-1}$; only 3 measurements out of 20 reach tens of µg g$^{-1}$ of Th (17, 18 and 38 µg g$^{-1}$ respectively). Pb contents are 9.4–27 µg g$^{-1}$, Pb/U ratios range

between 0.42–4.4 and µ values between 16–190.

Among the four samples selected for U–Pb geochronology, measured U contents of epidote are highly variable (2.6–530 µg g$^{-1}$; n = 80), and the intra-sample variability in U concentrations is ca. one order of magnitude in each sample (Fig. 3). Samples Albula-1 and Heyuan-1 overlap between a few to tens of µg g$^{-1}$ of U. Samples Grimsel-1 and Grimsel-2 overlap at U contents of hundreds of µg g$^{-1}$. Thorium concentrations span four orders of magnitude (0.009–38 µg g$^{-1}$; n = 56). Samples

with similar U concentrations display different Th contents, creating variability in Th/U ratios. Pb contents are 3.9–190 µg g$^{-1}$. Pb/U ratios span between 0.14–10, each sample varying to different extents. With only 4/88 Th measurements above 10 µg g$^{-1}$ (one in sample Albula-1 and three in sample Heyuan-1), Th–Pb geochronology is not viable. All epidote samples exhibit µ values well below 2000.

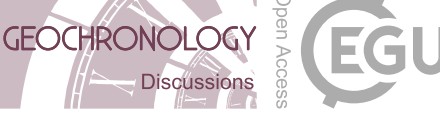

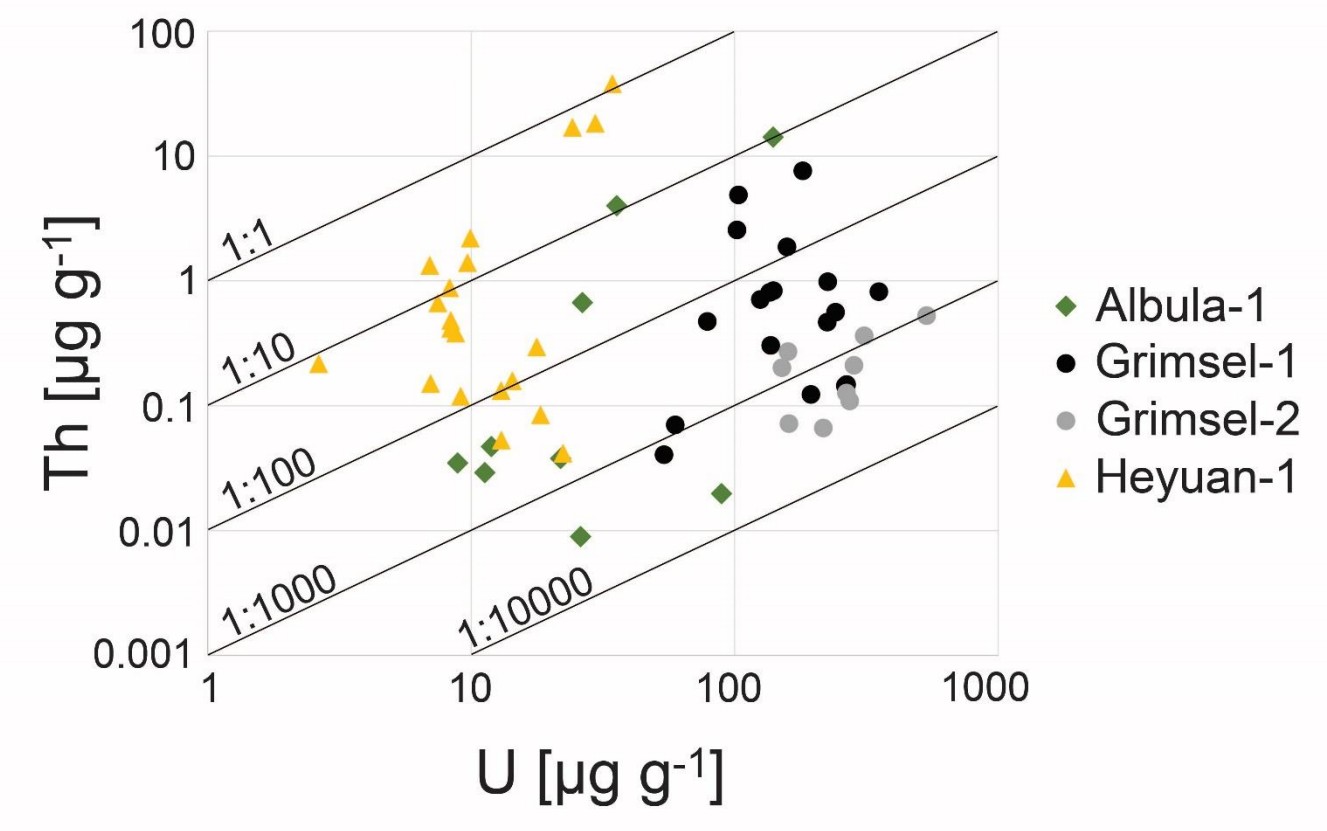

**Figure 3: Th and U contents of the analyzed epidote samples.**

**4.2 Testing Tara allanite as a reference material for epidote U–Pb geochronology**

To assess the validity of allanite as primary reference material for epidote dating, we compared the DF correction of $^{206}$Pb/$^{238}$U ratios over the ablation time using Tara allanite as reference for CAP[b] allanite (Burn et al., 2017) as matrix-matched, for Plesovice zircon (Sláma et al., 2008) as non-matrix-matched, and for epidote as closely matrix-matched (Figs. 4 and 5). An accurate correction for DF produces flat time-resolved lines of DF-corrected $^{206}$Pb/$^{238}$U ratios for unknowns: sloping or more complex shaped curves (Fig. 4b) indicate that the DF correction is not compensating for the difference in matrix. Parallel flat lines that plot below or above each other are indicative of variable initial Pb fractions, but still indicate proper correction for DF. As expected, CAP[b] allanite has DF-corrected $^{206}$Pb/$^{238}$U ratios that are flat when standardized to Tara allanite (Fig. 4a, both measured with a 50 μm spot). The DF corrected $^{238}$U/$^{206}$Pb ratios measured in Albula-1 and Grimsel-1 epidotes with 50 (Fig. 4c–d) and even 30 μm spots are flat (Fig. 5a–b). This indicates similar ablation behavior and downhole fractionation of U from Pb between epidote and allanite for our analytical setup, and accurate correction for DF in epidote by using Tara allanite as primary reference material. In contrast, the distinct ablation behavior of (Plesovice) zircon is revealed by complex shaped DF-corrected $^{238}$U/$^{206}$Pb ratios even with a 50 μm spot (Fig. 4b).





**Figure 4:** $^{206}Pb/^{238}U$ ratios measured by LA–ICP–MS corrected for downhole fractionation (DF) of (a) CAP[b] allanite, (b) Plesovice zircon, (c) Albula-1 and (d) Grimsel-1 epidote with Tara allanite as primary reference material. Measurements with a 50 μm spot size. The DF-corrected $^{206}Pb/^{238}U$ ratios include both initial and radiogenic Pb. Lines that are not overlapping in (c) and (d) indicate variable amounts of initial Pb in each measurement.





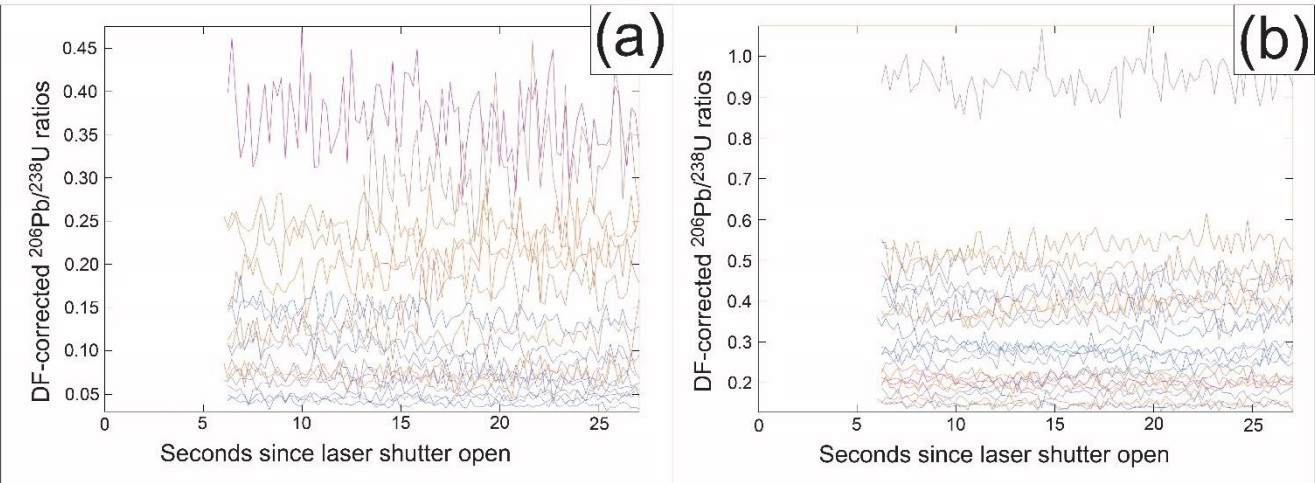

**Figure 5: $^{206}$Pb/$^{238}$U ratios measured by LA–ICP–MS corrected for downhole fractionation (DF) of (a) Albula-1 and (b) Grimsel-1 epidote with Tara allanite as primary reference material. Measurements with a 30 μm spot size. The DF-corrected $^{206}$Pb/$^{238}$U ratios include both initial and radiogenic Pb. Lines that are not overlapping indicate variable amounts of initial Pb in each measurement.**

**4.3 Laser ablation ICP–MS U–Pb data of unknown samples**

Uncertainties on LA–ICP–MS $^{238}$U/$^{206}$Pb and $^{207}$Pb/$^{206}$Pb ratios are 2 standard errors (2 SE), and all age uncertainties calculated with Isoplot are 95 % confidence.

**Table 5: $^{238}$U/$^{206}$Pb and $^{207}$Pb/$^{206}$Pb ratios and their uncertainties as 2 standard errors (2 SE) measured by LA–ICP–MS. Used spot size is indicated in brackets.**

| Sample | Analysis | $^{238}$U/$^{206}$Pb | 2 SE | $^{207}$Pb/$^{206}$Pb | 2 SE |
|---|---|---|---|---|---|
| Albula-1 | 1 | 12.05 | 0.33 | 0.734 | 0.015 |
| Vein1 | 2 | 8.237 | 0.258 | 0.765 | 0.017 |
| (50 μm) | 3 | 3.953 | 0.103 | 0.815 | 0.016 |
| | 4 | 0.535 | 0.013 | 0.8368 | 0.0067 |
| | 5 | 0.996 | 0.028 | 0.841 | 0.016 |
| | 6 | 0.952 | 0.025 | 0.820 | 0.013 |
| | 7 | 1.691 | 0.046 | 0.8225 | 0.0079 |
| | 8 | 1.707 | 0.041 | 0.8216 | 0.0098 |
| | 9 | 15.33 | 0.40 | 0.712 | 0.013 |
| | 10 | 4.024 | 0.097 | 0.7961 | 0.0069 |
| | 11 | 6.540 | 0.188 | 0.787 | 0.015 |
| | 12 | 11.53 | 0.29 | 0.743 | 0.015 |
| | 13 | 4.710 | 0.129 | 0.807 | 0.019 |
| | 14 | 2.747 | 0.128 | 0.802 | 0.013 |
| | 15 | 2.076 | 0.056 | 0.824 | 0.014 |




| | | | | | |
|---|---|---|---|---|---|
| | 16 | 14.77 | 0.39 | 0.718 | 0.016 |
| | 17 | 2.675 | 0.068 | 0.810 | 0.015 |
| | 18 | 6.053 | 0.253 | 0.776 | 0.014 |
| Albula-1 | 19 | 13.59 | 0.33 | 0.7332 | 0.0096 |
| Vein2 | 20 | 33.24 | 0.83 | 0.5858 | 0.0088 |
| (50 μm) | 21 | 18.46 | 0.44 | 0.689 | 0.0083 |
| | 22 | 8.576 | 0.213 | 0.7578 | 0.0094 |
| Albula-1 | 1 | 2.762 | 0.107 | 0.839 | 0.025 |
| Vein1 | 2 | 13.91 | 0.58 | 0.737 | 0.027 |
| 30 μm | 3 | 5.035 | 0.213 | 0.794 | 0.028 |
| | 4 | 4.102 | 0.143 | 0.788 | 0.013 |
| | 5 | 4.706 | 0.168 | 0.799 | 0.015 |
| | 6 | 8.569 | 0.338 | 0.763 | 0.028 |
| | 7 | 6.510 | 0.309 | 0.782 | 0.025 |
| | 8 | 3.236 | 0.157 | 0.778 | 0.038 |
| | 9 | 14.08 | 0.62 | 0.710 | 0.030 |
| | 10 | 13.37 | 0.59 | 0.735 | 0.032 |
| | 11 | 17.79 | 0.70 | 0.686 | 0.015 |
| | 12 | 21.90 | 0.81 | 0.639 | 0.017 |
| | 13 | 14.99 | 0.54 | 0.703 | 0.015 |
| | 14 | 10.54 | 0.77 | 0.740 | 0.017 |
| | 15 | 9.940 | 0.356 | 0.743 | 0.016 |
| | 16 | 21.70 | 0.80 | 0.658 | 0.018 |
| | 17 | 26.21 | 0.96 | 0.639 | 0.017 |
| | 18 | 7.107 | 0.263 | 0.776 | 0.013 |
| Grimsel-1 | 1 | 3.049 | 0.121 | 0.7864 | 0.0047 |
| (50 μm) | 2 | 2.593 | 0.059 | 0.7906 | 0.005 |
| | 3 | 4.929 | 0.114 | 0.7867 | 0.0062 |
| | 4 | 10.33 | 0.23 | 0.7738 | 0.0042 |
| | 5 | 7.896 | 0.181 | 0.7774 | 0.0058 |
| | 6 | 4.268 | 0.178 | 0.7888 | 0.0053 |
| | 7 | 2.443 | 0.055 | 0.7895 | 0.0061 |
| | 8 | 2.189 | 0.057 | 0.7887 | 0.0049 |
| | 9 | 3.615 | 0.081 | 0.7898 | 0.0054 |
| | 10 | 4.429 | 0.098 | 0.7867 | 0.0058 |
| | 11 | 2.789 | 0.062 | 0.7915 | 0.0056 |
| | 12 | 3.542 | 0.080 | 0.7896 | 0.0054 |
| | 13 | 4.647 | 0.119 | 0.785 | 0.0063 |
| | 14 | 1.098 | 0.024 | 0.7937 | 0.0042 |
| | 15 | 4.137 | 0.098 | 0.787 | 0.0045 |



|  |  |  |  |  |  |
|---|---|---|---|---|---|
|  | 16 | 6.826 | 0.158 | 0.7771 | 0.0048 |
|  | 17 | 2.912 | 0.067 | 0.7916 | 0.0045 |
|  | 18 | 5.258 | 0.127 | 0.7847 | 0.0048 |
|  | 19 | 4.737 | 0.112 | 0.7836 | 0.0058 |
|  | 20 | 6.449 | 0.145 | 0.7819 | 0.004 |
|  | 21 | 4.746 | 0.106 | 0.7941 | 0.0066 |
|  | 22 | 6.020 | 0.159 | 0.7843 | 0.0053 |
|  | 23 | 3.862 | 0.086 | 0.7877 | 0.0053 |
| Grimsel-1 | 1 | 5.018 | 0.166 | 0.7792 | 0.0096 |
| (30 µm) | 2 | 2.533 | 0.090 | 0.7839 | 0.0094 |
|  | 3 | 6.575 | 0.220 | 0.785 | 0.011 |
|  | 4 | 2.674 | 0.086 | 0.7852 | 0.0089 |
|  | 5 | 5.266 | 0.180 | 0.763 | 0.012 |
|  | 6 | 2.089 | 0.070 | 0.7777 | 0.0074 |
|  | 7 | 4.640 | 0.155 | 0.783 | 0.011 |
|  | 8 | 1.895 | 0.065 | 0.7855 | 0.0091 |
|  | 9 | 5.339 | 0.177 | 0.7667 | 0.0085 |
|  | 10 | 4.684 | 0.158 | 0.781 | 0.0086 |
|  | 11 | 6.973 | 0.233 | 0.780 | 0.011 |
|  | 12 | 2.123 | 0.077 | 0.7867 | 0.0085 |
|  | 13 | 1.077 | 0.036 | 0.7842 | 0.007 |
|  | 14 | 2.168 | 0.070 | 0.783 | 0.01 |
|  | 15 | 2.913 | 0.093 | 0.7794 | 0.0085 |
|  | 16 | 2.461 | 0.079 | 0.7769 | 0.009 |
|  | 17 | 2.685 | 0.086 | 0.7874 | 0.009 |
|  | 18 | 2.357 | 0.078 | 0.7771 | 0.0098 |
|  | 19 | 3.840 | 0.131 | 0.777 | 0.010 |
|  | 20 | 7.032 | 0.232 | 0.7703 | 0.0095 |
|  | 21 | 5.862 | 0.196 | 0.7687 | 0.0099 |
|  | 22 | 3.611 | 0.117 | 0.7763 | 0.0065 |
|  | 23 | 4.572 | 0.178 | 0.7781 | 0.0099 |
|  | 24 | 3.636 | 0.132 | 0.7779 | 0.0090 |
|  | 25 | 3.347 | 0.123 | 0.7804 | 0.0080 |
| Grimsel-2 | 1 | 13.83 | 0.31 | 0.7722 | 0.0058 |
| (50 µm) | 2 | 19.11 | 0.44 | 0.7619 | 0.0059 |
|  | 3 | 13.34 | 0.30 | 0.7728 | 0.0061 |
|  | 4 | 8.929 | 0.215 | 0.7841 | 0.0055 |
|  | 5 | 12.56 | 0.284 | 0.7741 | 0.0066 |
|  | 6 | 11.88 | 0.27 | 0.7802 | 0.0069 |
|  | 7 | 8.376 | 0.189 | 0.7832 | 0.0068 |





|  | | | | |
|---|---|---|---|---|
|  | 8 | 8.547 | 0.248 | 0.7843 | 0.0061 |
|  | 9 | 13.93 | 0.43 | 0.7734 | 0.0062 |
|  | 10 | 10.37 | 0.29 | 0.7791 | 0.0061 |
|  | 11 | 8.353 | 0.188 | 0.7812 | 0.0060 |
|  | 12 | 9.346 | 0.262 | 0.7825 | 0.0068 |
|  | 13 | 20.07 | 0.48 | 0.7587 | 0.0070 |
|  | 14 | 9.329 | 0.218 | 0.7811 | 0.0065 |
|  | 15 | 8.765 | 0.200 | 0.7774 | 0.0063 |
|  | 16 | 10.34 | 0.23 | 0.7797 | 0.0071 |
| Heyuan-1 | 1 | 5.040 | 0.155 | 0.764 | 0.016 |
| (50 µm) | 2 | 3.565 | 0.108 | 0.772 | 0.012 |
|  | 3 | 5.907 | 0.174 | 0.734 | 0.017 |
|  | 4 | 4.958 | 0.160 | 0.757 | 0.012 |
|  | 5 | 2.655 | 0.071 | 0.785 | 0.013 |
|  | 6 | 5.244 | 0.187 | 0.742 | 0.012 |
|  | 7 | 3.499 | 0.098 | 0.764 | 0.016 |
|  | 8 | 6.676 | 0.209 | 0.716 | 0.019 |
|  | 9 | 4.409 | 0.132 | 0.757 | 0.013 |
|  | 10 | 2.793 | 0.117 | 0.777 | 0.014 |
|  | 11 | 9.578 | 0.349 | 0.678 | 0.016 |
|  | 12 | 3.772 | 0.119 | 0.775 | 0.016 |
|  | 13 | 4.751 | 0.165 | 0.759 | 0.016 |
|  | 14 | 5.525 | 0.217 | 0.749 | 0.019 |
|  | 15 | 5.827 | 0.180 | 0.739 | 0.017 |
|  | 16 | 1.513 | 0.050 | 0.797 | 0.016 |
|  | 17 | 2.252 | 0.081 | 0.794 | 0.015 |
|  | 18 | 1.570 | 0.052 | 0.801 | 0.016 |
|  | 19 | 18.25 | 0.57 | 0.596 | 0.02 |
|  | 20 | 1.595 | 0.064 | 0.795 | 0.019 |
|  | 21 | 2.453 | 0.078 | 0.775 | 0.017 |
|  | 22 | 0.753 | 0.021 | 0.805 | 0.014 |
|  | 23 | 1.435 | 0.047 | 0.809 | 0.016 |

A total of twenty-two spot analyses were measured in sample Albula-1 with a 50 µm spot. $^{238}U/^{206}Pb$ ratios are 0.535–33.2 with uncertainties of 2–4 %. $^{207}Pb/^{206}Pb$ ratios are 0.586–0.837 ± 0.8–2 % (Table 5). A Tera–Wasserburg regression based on all data points (Fig. 6a) yields a lower concordia intercept age of 62.7 ± 3.0 Ma (MSWD = 1.6) with an upper $^{207}Pb/^{206}Pb$ intercept of 0.8334 ± 0.0043. The good spread of the data points along the regression line reflects variable fractions of initial Pb. The small MSWD value indicates that there is no resolvable age difference between the two veins at the present

analytical precision. In fact, if the analyses from the two veins are considered separately, the Tera–Wasserburg ages of Vein1





and Vein2 are 67.6 ± 5.0 Ma (n = 18) and 58.9 ± 3.8 Ma (n = 4), which overlap within uncertainty. The initial $^{207}Pb/^{206}Pb$ ratio of Albula-1 epidote indicated by the Tera-Wasserburg diagram is within uncertainty of the modeled value of initial $^{207}Pb/^{206}Pb$ of 0.840 ± 0.015 at 63 Ma (Stacey & Kramers, 1975). By using this modeled initial ratio for a $^{207}Pb$-correction, the weighted average $^{207}Pb$-corrected $^{238}U/^{206}Pb$ age is 65.0 ± 2.5 Ma (MSWD = 0.91). Eighteen additional measurements

were carried out in sample Albula-1 (10 in Vein1 and eight in Vein2) with a spot size of 30 μm (Table 5). The Tera–Wasserburg diagram based on all 30 μm analyses yields an intercept age of 65.4 ± 4.6 Ma and a $^{207}Pb/^{206}Pb$ intercept of 0.8297 ± 0.0086 (MSWD = 1.4). Both values are within uncertainty of those obtained with a spot size of 50 μm.

**Figure 6: Tera–Wasserburg diagrams of (a) Albula-1, (b) Grimsel-1, (b) Grimsel-2 and (d) Heyuan-1 epidote with 50 μm**
**measurements. Ages are calculated from the lower intercept of the regressions through the analyses with the concordia, whereas initial $^{207}Pb/^{206}Pb$ are calculated from the upper intercept of the regressions with the y-axis. Data-point error ellipses are 2σ and age uncertainties are 95 % confidence.**





Twenty-three 50 μm analyses were carried out in sample Grimsel-1 and define a regression in a Tera–Wasserburg diagram (Fig. 6b) with an intercept age of 19.1 ± 4.0 Ma (MSWD = 0.75) and a $^{207}$Pb/$^{206}$Pb intercept of 0.7963 ± 0.0024. All data

points plot close to the y-axis ($^{238}$U/$^{206}$Pb ratios = 1.10–10.3 ± 2–4 %; $^{207}$Pb/$^{206}$Pb ratios = 0.774–0.794 ± 0.5–0.8 %; Table 5), indicating high and poorly variable initial Pb fractions in all measurements. The initial $^{207}$Pb/$^{206}$Pb value is outside uncertainty of the modeled initial $^{207}$Pb/$^{206}$Pb ratio of 0.837 ± 0.015 at 19 Ma of Stacey and Kramers (1975). Hence, a $^{207}$Pb-correction with the modeled ratio would yield an inaccurate weighted average $^{207}$Pb-corrected $^{238}$U/$^{206}$Pb age; thus this sample can only be dated by the Tera–Wasserburg approach. Twenty-five additional measurements were made with a spot

size of 30 μm (Table 5). Four data points are rejected as outliers as they cause higher MSWDs and lower age precision if included in the Tera–Wasserburg diagram. The regression based on 21 analyses intercepts the concordia at 24 ± 11 Ma (MSWD = 0.80) with a $^{207}$Pb/$^{206}$Pb intercept of 0.7884 ± 0.0047. The age remains within uncertainty of that calculated with a 50 μm spot size.

Sixteen 50 μm analyses of sample Grimsel-2 define a regression in the Tera–Wasserburg diagram (Fig. 6c) yielding an age

of 16.9 ± 3.7 Ma (MSWD = 0.40). The spread along the regression is poor and the data points are close to the y-axis ($^{238}$U/$^{206}$Pb ratios = 8.35–20.1 ± 2–3 %; $^{207}$Pb/$^{206}$Pb ratios = 0.759–0.784 ± 0.7–0.9 %; Table 5), indicating high and similar fractions of initial Pb in all analyses. The initial $^{207}$Pb/$^{206}$Pb ratio of 0.7998 ± 0.0054 is outside uncertainty of the modeled initial $^{207}$Pb/$^{206}$Pb value of 0.837 ± 0.015 at 17 Ma (Stacey and Kramers, 1975), which cannot be used for a $^{207}$Pb-correction. The Tera–Wasserburg approach is therefore the only viable to date this sample.

Twenty-three analyses were carried out in sample Heyuan-1 with a spot size of 50 μm, defining a regression (Fig. 6d) with an age of 108.1 ± 8.4 Ma (MSWD = 0.92). Initial Pb fractions are highly variable in the different measurements yielding appreciable spread along the regression ($^{238}$U/$^{206}$Pb ratios = 0.753–18.2 ± 3–4 %; $^{207}$Pb/$^{206}$Pb ratios = 0.596–0.809 ± 2–3 %; Table 5). The initial $^{207}$Pb/$^{206}$Pb ratio of 0.8170 ± 0.0055 indicated by the upper intercept is outside uncertainty of the modeled initial $^{207}$Pb/$^{206}$Pb value of 0.843 ± 0.015 at 108 Ma (Stacey and Kramers, 1975). Thus, the age of this sample can

only be determined from the Tera-Wasserburg diagram.

On a final note, we have calculated Tera–Wasserburg ages of the presented epidote samples by using CAP[b] (June 2019; reference values from Burn et al., 2017) and CAP (July 2019 and January 2020; reference values from Barth et al., 1994) allanite as primary reference materials to assess if the calculated epidote U–Pb ages change when implementing a different allanite as primary reference material. The resulting epidote U–Pb ages remain within uncertainty of those calculated with

Tara allanite (Appendix C).

### 4.4 Solution ICP–MS U–Pb data

Uncertainties on solution ICP–MS $^{238}$U/$^{206}$Pb and $^{207}$Pb/$^{206}$Pb ratios are 2 standard errors (2 SE).

**Table 6: $^{238}$U/$^{206}$Pb and $^{207}$Pb/$^{206}$Pb ratios measured by solution ICP–MS. Uncertainties are given as 2 standard errors (2 SE).**



| Sample | $^{238}$U/$^{206}$Pb | 2 SE | $^{207}$Pb/$^{206}$Pb | 2 SE |
|---|---|---|---|---|
| Albula-1_A | 3.67 | 0.03 | 0.81319 | 0.00004 |
| Albula-1_B | 3.61 | 0.04 | 0.81337 | 0.00003 |
| Albula-1_C | 3.04 | 0.04 | 0.81674 | 0.00003 |
| Albula-1_D | 3.08 | 0.05 | 0.81674 | 0.00003 |


For measurements by solution ICP–MS of $^{238}$U/$^{206}$Pb and $^{207}$Pb/$^{206}$Pb ratios in Albula-1 epidote, ca. 30 mg of material were necessary to ensure ca. 300 ng of total Pb. $^{238}$U/$^{206}$Pb ratios range between 3.04–3.67 with uncertainties between 0.8–1.6 %, and $^{207}$Pb/$^{206}$Pb ratios between 0.81319–0.81674 ± 0.03–0.04 ‰ (Table 6). The uncertainties on solution ICP–MS $^{238}$U/$^{206}$Pb ratios are lower than those measured by LA–ICP–MS with a 50 µm spot size by a factor of 2.5, and a decrease by a factor of

13–100 occurs in analytical uncertainties on $^{207}$Pb/$^{206}$Pb ratios. The two aliquots from each epidote micro-separate (A–B and C–D in Table 6) yield identical ratios within uncertainty. $^{238}$U/$^{206}$Pb and $^{207}$Pb/$^{206}$Pb ratios display meager spread in a Tera–Wasserburg diagram (Fig. 7). In comparison to LA–ICP–MS data, the intra-sample variability of the solution ICP–MS $^{238}$U/$^{206}$Pb and $^{207}$Pb/$^{206}$Pb ratios is only 2 % and 14 % respectively, attesting to homogenization of initial Pb fractions in the micro-separates. This confirms that no statistically robust Tera-Wasserburg regression can be calculated from the solution

ICP–MS data alone. The $^{238}$U/$^{206}$Pb and $^{207}$Pb/$^{206}$Pb ratios measured by solution ICP–MS overlap with individual LA–ICP–MS data points within their error envelope in a Tera–Wasserburg diagram (50 µm spot; Fig. 7).



**Figure 7: Tera–Wasserburg diagram showing the comparison between laser ablation ICP–MS and solution ICP–MS data points. The error envelope of LA–ICP–MS data points is calculated with a MatLab script and is based on the scatter of the data points.**
**Data-point error ellipses are 2σ.**

## 5 Discussion

### 5.1 CAP[b], CAP and AVC allanite as quality control

Although allanite geochronology is beyond the scope of this contribution, analyses of CAP[b], CAP and AVC allanite provide a quality control on U–Pb measurements during the analytical sessions. Since the epidote samples analyzed in this study
contain too little Th for Th–Pb geochronology and we only explore epidote as a U–Pb geochronometer, we compare our measured allanite data to published U–Pb ages rather than their Th–Pb reference ages because ages from the two systems





differ (Barth et al., 1994; Gregory et al., 2007; 2012; Darling et al., 2012; El Korh, 2014; Smye et al., 2014; Burn et al., 2017) due to open-system behavior of the U–Pb system as opposed to the Th–Pb one (Barth et al., 1994). Tera–Wasserburg and weighted average $^{207}$Pb-corrected $^{238}$U/$^{206}$Pb ages determined from our analyses of CAP$^b$, CAP and AVC allanite (Table

3) in all analytical sessions are overall consistent with published U–Pb ages, attesting to reliable U–Pb measurements. AVC allanite analyzed in two sessions provided Tera–Wasserburg ages of 292.4 ± 3.7 Ma and 285.2 ± 4.5 Ma and identical weighted average $^{207}$Pb-corrected $^{238}$U/$^{206}$Pb ages. Our AVC allanite ages are within uncertainty of the Tera–Wasserburg age of 289.6 ± 5.6 Ma of Gregory et al. (2007). CAP$^b$ allanite (Burn et al., 2017) was used in one session, yielding a Tera–Wasserburg age of 284.2 ± 2.6 Ma and a weighted average $^{207}$Pb-corrected $^{238}$U/$^{206}$Pb age of 284.2 ± 2.0 Ma, both within

uncertainty of those calculated by Burn et al. (2017) of respectively 284.9 ± 2.8 Ma and 283.8 ± 2.8 Ma. CAP allanite was used in two sessions returning intercept ages of 288.5 ± 2.9 Ma and 283.0 ± 3.4 Ma, with comparable weighted average $^{207}$Pb-corrected $^{238}$U/$^{206}$Pb ages. Our CAP U–Pb ages are within uncertainty of or close to the U–Pb age of 275.0 ± 4.7 Ma determined by SHRIMP by Gregory et al. (2007). The ages of CAP$^b$, CAP and AVC allanite obtained from unanchored Tera-Wasserburg regressions are identical to those obtained from the anchored regressions but less precise.

The secondary reference materials are also used to calculate the systematic uncertainty of analyses, which should be propagated with the age uncertainty of the samples (see Fig. 1 in Horstwood, 2016). Since no long-term repeated measurements of U–Pb ages of our allanite secondary reference materials are available on the used LA–ICP–MS system, this step cannot be done for our epidote samples. However, a worst-case scenario can be envisioned by considering a systematic uncertainty of 5 %, which is the difference between the Tera–Wasserburg age measured in CAP during the July 2019 session

and its reference Th–Pb age. The age and its propagated 2 SE would be 62.7 ± 4.3 Ma for Albula-1 epidote, 19.1 ± 4.1 Ma for Grimsel-1 epidote, 16.9 ± 3.8 Ma for Grimsel-2 epidote and 108.1 ± 10 Ma for Heyuan-1 epidote. Therefore, even if the difference of 5 % between the CAP allanite age in the July 2019 session and the reference U–Pb age of Gregory et al. (2007) could be regarded as systematic uncertainty, it would not have a significant effect on our epidote U–Pb age uncertainties.

**5.2 Tara allanite as reference material for LA–ICP–MS standardization of epidote**

The presented data confirm that Tara allanite is an appropriate primary reference material for U–Pb dating of epidote by LA–ICP–MS in spot-analysis mode. The primary reference material is used to normalize the measured isotopic ratios to real values after correcting them for DF, which is crucial to obtain reliable U–Pb geochronology by LA–ICP–MS (Horstwood et al., 2016). In this respect, the DF-corrected $^{206}$Pb/$^{238}$U ratios of epidote display a flat trend throughout ablation time, demonstrating that the correction for DF is accurately carried out by using Tara allanite with a spot of 50 μm (Fig. 4c–d) and

even 30 μm (Fig. 5). This is corroborated by the fact that the Tera–Wasserburg ages and initial $^{207}$Pb/$^{206}$Pb ratios of Albula-1 and Grimsel-1 epidotes with a 30 μm spot remain consistent with the dataset with a 50 μm spot. However, in both epidote samples age precision decreases with a spot of 30 μm. In the case of Grimsel-1 epidote, this is expected because the poor spread in $^{238}$U/$^{206}$Pb and $^{207}$Pb/$^{206}$Pb ratios gets combined with larger analytical uncertainties, leading to an even more poorly





constrained regression than that with a 50 μm spot size. Thanks to the larger spread in plotted ratios of Albula-1 epidote, the
effects of the lower analytical precision with a 30 μm spot size are less dramatic but still noticeable.

## 5.3 Validation of $^{238}$U/$^{206}$Pb and $^{207}$Pb/$^{206}$Pb ratios by solution ICP–MS and considerations on analyzed volumes versus age precision

The consistency between the differently acquired datasets lands support to the accuracy of our LA–ICP–MS data and of the calculated Tera–Wasserburg ages. The much more limited variability of $^{238}$U/$^{206}$Pb ratios measured by solution ICP–MS data
with respect to LA–ICP–MS acquisitions is expected because sample preparation for solution ICP–MS requires homogenization of ca. $9 \times 10^9$ μm$^3$ (ca. 30 mg of material). By comparison, with LA–ICP–MS homogenization occurred over a volume of ca. $20$–$24 \times 10^3$ μm$^3$, as the measured depth of the LA–ICP–MS craters is between 10–12 μm. In fact, the intra-sample variability of solution ICP–MS data is limited to the extent that the two aliquots from each epidote micro-separate yield indistinguishable $^{238}$U/$^{206}$Pb and $^{207}$Pb/$^{206}$Pb ratios despite the high analytical precision.

With decreasing spread in the plotted $^{238}$U/$^{206}$Pb and $^{207}$Pb/$^{206}$Pb ratios, the extrapolation of the lower intercept of the regression with the concordia becomes less constrained. Epidote samples Albula-1 and Heyuan-1 display the largest spreads in LA–ICP–MS data points and their age uncertainties are 5 % and 8 % respectively. Analyses in Grimsel-1 and Grimsel-2 epidote define small spreads and give large uncertainties on the Tera–Wasserburg ages (21 % and 22 % respectively). Notably, samples Grimsel-1 and Grimsel-2 have the highest U contents (hundreds of μg g$^{-1}$ versus tens of μg g$^{-1}$ in samples
Albula-1 and Heyuan-1) and hence the best counting statistics. This suggests that higher analytical precision alone does not ensure the best possible age precision unless it is accompanied by a large-enough data-point spread. It also confirms that epidote samples with relatively low U contents should not be automatically considered unsuitable for U–Pb geochronology.

From the considerations on analyzed volumes by both LA– and solution ICP–MS versus data-point spread in Tera–Wasserburg diagrams, combined with those on age uncertainties and statistical robustness of regressions, it is crucial that the
size of the analyses represent a compromise ensuring high-enough analytical precision and the smallest extent possible of sample homogenization. As a result, the variability in initial Pb fractions is preserved, which is fundamental in determining large-enough spread along the Tera–Wasserburg regression and higher age precision. It should be noted that the analyzed samples are characterized by variable Th/U ratios (Fig. 2). The fractionation of Th from U is commonly attributed to oxidizing conditions (e.g. Frei et al., 2004). Although at the current state of research we cannot draw conclusions in this
respect, it is possible that the variability in initial Pb fractions of each individual epidote vein might be determined by physico-chemical conditions upon epidote crystallization, such as the above-mentioned oxidizing conditions or re-equilibration along fluid pathways.

## 5.4 Isotopic composition of initial Pb

Among the epidote samples analyzed in this study, only Albula-1 epidote gives an initial $^{207}$Pb/$^{206}$Pb ratio that overlaps
within uncertainty with the modeled value of Stacey & Kramers (1975). Accordingly, its weighted average $^{207}$Pb-corrected



$^{238}U/^{206}Pb$ age is within uncertainty of its Tera–Wasserburg age. Epidotes from all other samples yielded Tera–Wasserburg initial $^{207}Pb/^{206}Pb$ ratios that deviate from modeled values, indicating non-negligible additions of radiogenic components to the initial Pb. Radiogenic Pb components can be inherited by the fluid at its source and/or during circulation and re-equilibration with rocks along its pathway containing U-Th-bearing minerals (e.g. Romer, 2001). The weighted average

$^{207}Pb$-corrected $^{238}U/^{206}Pb$ ages of epidote Grimsel-1, Grimsel-2 and Heyuan-1 calculated by assuming a modeled initial $^{207}Pb/^{206}Pb$ ratio would be grossly inaccurate, implying that these three samples can only be dated with a Tera–Wasserburg diagram. These considerations confirm the Tera–Wasserburg approach as the most suitable – and often the only viable – for accurate U–Pb dating of low-μ phases such as epidote (see Romer, 2001; Romer and Xiao, 2005).

## 5.5 Geological constraints on epidote U–Pb ages

To evaluate the geological accuracy of the U–Pb ages calculated from epidote in the veins presented above we consider other geochronological constraints on the deformation history of their respective host rocks. Albula-1 epidote gives a Paleocene age of 62.7 ± 3.0 Ma. Although, to our knowledge, no isotope geochronology is available in the Albula region, our epidote U–Pb age is consistent with geodynamic events taking place in its surroundings. For example, the Err–Platta system was investigated by Handy et al. (1996) and their D2 – in the stability field of epidote – is dated 80–67 Ma (K–Ar on white

mica); epidote growth is also observed in the post-D2 deformation (Handy et al., 1996). A rutile U–Pb age of 63.0 ± 3.0 Ma was calculated by Picazo et al. (2019) from the Malenco–Margna boundary (Passo d'Ur, ca. 90 km south-southeast of the Albula area; Fig. 1a in Picazo et al., 2019) dating the stacking of the nappes associated with metamorphism at high pressure. Epidote U–Pb ages in samples Grimsel-1 and Grimsel-2 yield (early) Miocene ages of 19.1 ± 4.0 Ma and 16.9 ± 3.7 Ma respectively. These ages are within uncertainty of each other and can be attributed to the early deformation in the area

between 22–17 Ma (Handegg phase; Challandes et al., 2008; Rolland et al., 2009). This is corroborated by the presence of green biotite associated with epidote in the epidote-bearing veins (Challandes et al., 2008; Rolland et al., 2009; Herwegh et al., 2017; Wehrens et al., 2017). Notably, Grimsel-1 and Grimsel-2 epidote samples yield initial $^{207}Pb/^{206}Pb$ ratios that are identical within uncertainty and indicate an inherited radiogenic component. This implies that the Pb isotopic signature of the circulating fluid(s) was homogenously re-equilibrated over a ca. 200 m distance.

Epidote of sample Heyuan-1 yields an age of 108.1 ± 8.4 Ma, which is (early) Cretaceous. Since the sample is crosscut by an earliest-generation quartz vein associated with hematite (Tannock et al., 2020a; 2020b), this age is consistent with the earliest movements of the Heyuan Fault.

## 5.6 Epidote ages as time of crystallization in low-temperature veins

Having established that the calculated epidote U–Pb ages are consistent with geological events that affected the host rocks,

we now discuss whether these ages can be truly considered as representative of epidote crystallization. The highest temperatures recorded by the deformation events that affected the meta-granitoid rocks hosting the analyzed epidote veins at Albula Pass, at Grimsel Pass and at the Heyuan Fault are respectively 300 °C (Mohn et al., 2011), 450 ± 30 °C (Challandes



et al., 2008; Goncalves et al., 2012) and 330 °C (Tannock et al., 2020a). All these temperatures are well below 685–750 °C,
proposed by Dahl (1997) as the range for the closure temperature of Pb diffusion in epidote. Nevertheless, resetting of the

U–Pb geochronometer can occur independently of temperature via fluid-mediated dissolution–precipitation processes, which
can be assessed with BSE imaging. Albula-1 and Grimsel-2 epidotes display growth zoning, which is regarded as primary
zoning and thus lack of significant elemental diffusion (Franz and Liebscher, 2004). Since care was taken to avoid mixing of
different zoning domains in each single analysis – including those associated with secondary veinlets – and considering that
the MSWDs of the calculated epidote ages are all close to or below 1 (i.e. only one epidote generation can be distinguished

at the current analytical precision), we can conclude that the ages of Albula-1 and Grimsel-2 epidote date their crystallization
and therefore the formation of the epidote-bearing veins.

The epidote-bearing vein in sample Grimsel-1 is folded; epidote does not display prominent zoning, and it is fractured and
porous. This may raise questions as to whether 19.1 ± 4.0 Ma represents the formation of Grimsel-1 epidote or the complete
resetting of the U–Pb geochronometer by interaction with a fluid assisting the Alpine deformation. However, epidote is

associated with green biotite and chlorite is locally present, hinting that the vein epidote formed at 19.1 ± 4.0 Ma during the
Handegg phase in the stability field of green biotite (Challandes et al., 2008; Rolland et al., 2009; Herwegh et al., 2017;
Wehrens et al., 2017). The subsequent folding of the vein may have occurred at the end of the Handegg phase or at the
beginning of the Oberaar phase with the onset of chlorite crystallization (Herwegh et al., 2017; Wehrens et al., 2017).

In sample Heyuan-1, epidote is present in pockets filled by an epidote-quartz(-chlorite) assemblage. This might suggest a

magmatic origin of epidote and consequently that epidote might have formed in the Jurassic as a magmatic mineral and that
the U–Pb system was reset by ingression of fluids related to the first movements of the Heyuan Fault. However, a magmatic
origin of the epidote can be ruled out based on the association of epidote with chlorite instead of biotite (the magmatic
phyllosilicate stable in the Xinfengjiang pluton; Li et al., 2007; Tannock et al., 2020a; 2020b), the former also consistent
with the temperature of mylonitization (330 °C; Tannock et al., 2020a). Furthermore, the Th/U ratios measured in Heyuan-1

epidote are << 1, whereas the Fogang Batholith – which comprises the Xinfengjiang pluton – has Th/U ratios >> 1 (Li et al.,
2007). We thus conclude that all epidote U-Pb ages presented in this study date the crystallization of the epidote grains that
formed during low-temperature fluid circulation.

## 6 Concluding remarks and future prospects

This contribution presents a protocol to obtain U–Pb ages and initial $^{207}$Pb/$^{206}$Pb compositions from monoclinic epidote, a

mineral highly but variably enriched in initial Pb. This includes preliminary screening of the material to verify the presence
of sufficiently high U contents (mainly between 7–310 μg g$^{-1}$ in our samples) and intra-sample chemical variability. If these
geochemical characteristics are ascertained, measurements by spot-analysis LA–ICP–MS using a quadrupole mass
spectrometer can allow U–Pb ages to be determined with uncertainties between ca. 5–20 %, the lowest precision being
related to poor variability in initial Pb fractions. It is demonstrated that epidote and allanite have similar downhole





fractionation of Pb from U during LA–ICP–MS spot analysis, and the consistency between the data measured by LA–ICP–MS and solution ICP–MS corroborates the accuracy of $^{238}U/^{206}Pb$ and $^{207}Pb/^{206}Pb$ ratios determined by using Tara allanite as primary reference material. We have shown that all effects due to downhole fractionation are accurately corrected for even with a spot size as small as 30 μm by using Tara allanite as primary reference material. Therefore, the lack of a standard that is perfectly matrix-matched to epidote does not prevent U–Pb dating of monoclinic epidote by spot-analysis LA–ICP–MS

with precision between ca. 5–20 %. The reliability of age data is also verified by the fact that they fit well in the geological evolution of the areas of origin of the epidote samples, and it has been demonstrated that the presented epidote ages date epidote crystallization.

    The key strategy for U–Pb dating of epidote is the Tera–Wasserburg diagram. U–Pb geochronology of epidote is most successful when the epidote samples display large-enough variability in initial Pb fractions – even when high analytical

precision is achieved – which may be related to variable physico-chemical conditions during the crystallization of vein-filling epidote. Although it is recommended that the largest spot size possible be used to ensure good counting statistics, it is imperative that geochemical heterogeneity be preserved among the different analyses in order to obtain a well-constrained Tera–Wasserburg regression and a small age uncertainty. An unexpected perk highlighted by the present study is that relatively low U contents (i.e. tens of μg g$^{-1}$) do not necessarily hamper age determinations at a geologically useful precision

provided that the spread of $^{238}U/^{206}Pb$ and $^{207}Pb/^{206}Pb$ ratios is large enough.

    This study presents a protocol that can be readily applied to date epidote-bearing hydrothermal veins and to assess initial Pb isotopic variability of the epidote-forming fluid. Better insight can now be gained from the application of epidote U–Pb dating into the mechanisms that led to the hydration of the continental crust in the Aar Massif and in the Err nappe. This study represents the base from which further developments may allow to date high-P epidote-bearing veins in subducted

oceanic units and to determine at the same time where the vein-forming fluid originated from thanks to the combination of trace element, age and isotopic data measured in epidote. Multiple phases of fault re-activation may be identified in fault-plane epidote. Whether or not plagioclase recrystallization in metamorphosed granitoid rocks is linked to the formation of epidote-bearing veins may be solved by measuring U–Pb ages and initial Pb isotopic compositions in epidote, which has proven its potential to become an invaluable geochemical, isotopic and geochronological tool.

**Appendix A**

    U, Th and Pb contents of epidote samples were measured on thin (30 μm) or thick (50–60 μm) sections, pre-cleaned with ethanol. Concentrations in sample Albula-1 were measured on a Geolas Pro 193 nm ArF excimer laser (Coherent, USA) coupled with an ELAN DRCe quadrupole ICP–MS (QMS; Perkin Elmer, USA). Instrument optimization and measurement procedures [similar to those reported in Pettke et al. (2012)] employed an ablation rate of ca. 0.1 μm per laser pulse, 10 Hz,

and beam sizes of between 24 and 60 μm to minimize limits of detection and to avoid inclusions and fractures. Ablation was done in a 1 l min$^{-1}$ He - 0.008 l min$^{-1}$ H$_2$ atmosphere. Concentrations in samples Grimsel-1, Grimsel-2 and Heyuan-1 were



measured on a RESOlutionSE 193 nm excimer laser system (Applied Spectra, USA) equipped with a S-155 large-volume constant-geometry chamber (Laurin Technic, Australia) coupled with an Agilent 7900 ICP–QMS. Ablation was carried out in He atmosphere, which was allowed to mix with Ar carrier gas for transport to the ICP–MS. Repetition rate was 5 Hz, at spot sizes between 20 and 50 μm.

On both systems, analytical conditions were optimized on NIST SRM612 so as to keep ThO production rate < 0.2 % and Th/U sensitivity ratio of 0.97–1.0, the latter indicative of robust plasma conditions. GSD-1G from USGS was employed as the external standard whereas quality control was monitored by measuring SRM612 from NIST measured as an unknown. A true-time linear drift correction was applied by bracketing standardization. Data acquired on both systems were reduced off-line using SILLS (Guillong et al., 2008), with the sum of measured total oxides (98.3 % for epidote and 100 % for SRM-NIST 612) used for internal standardization (compare Halter et al., 2002). Limits of detection were rigorously calculated for each element in each analysis employing the formulation detailed in Pettke et al. (2012).

**Appendix B**

Tera–Wasserburg diagrams of allanite secondary reference materials. All regressions are anchored to a $^{207}Pb/^{206}Pb$ intercept of 0.854 ± 0.015 at 275 Ma following Stacey and Kramers (1975) and all age uncertainties are 95 % confidence.





**Figure B1: Tera–Wasserburg diagrams of allanite secondary standards. (a) CAP[b] allanite, 14 June 2019 session; (b) CAP allanite, 23 July 2019 session; (c) CAP allanite, 16 January 2020 session. (d) AVC allanite, 23 July 2019 session; (e) AVC allanite, 16 January 2020 session. Data-point error ellipses are 2σ and age uncertainties are 95 % confidence.**

## Appendix C

**Table C1: U–Pb ages of epidote samples with CAP[b] (14 June 2019) and CAP (23 July 2019) allanites as primary reference materials. Age uncertainties are 95 % confidence.**

| Sample | 14 June 2019 | 23 July 2019 |
|---|---|---|
| | Tera–Wasserburg U–Pb age [Ma] | Tera–Wasserburg U–Pb age [Ma] |
| Albula-1 | 61.1 ± 2.8 MSWD = 1.6 n = 22 | - |
| Grimsel-1 | 18.7 ± 3.9 MSWD = 0.74 n = 23 | - |
| Grimsel-2 | 16.5 ± 3.5 MSWD = 0.39 n = 16 | - |
| Heyuan-1 | - | 102.5 ± 8.4 MSWD = 0.87 n = 23 |

**Author contribution**

Tanya Ewing, Daniela Rubatto and Alfons Berger were involved in planning and running LA–ICP–MS U–Pb dating, and provided supervision during all steps of the study. Martin Wille and Igor Villa supervised the work in the clean lab for digestion of epidote micro-separates, as well as the measurements and data reduction of solution ICP–MS data. Pierre Lanari contributed to the discussions on the analytical LA–ICP–MS setup and provided fundamental insight into LA–ICP–MS data evaluation and processing. Thomas Pettke enabled the measurements of Pb, Th and U contents by LA–ICP–MS. Marco Herwegh helped with the geological interpretation of the data. All authors were involved in data evaluation and interpretation, and contributed to the improvement of the manuscript. Veronica Peverelli participated in every step of the study and prepared the manuscript.

**Acknowledgements**

Veronica Peverelli would like to thank Francesca Piccoli, Gabriela Baltzer and Daniel Rufer for the help in the laboratories, Lisa Tannock for providing samples and Patrick Neuhaus from the Geography Department of University of Bern for carrying out the measurements of $^{238}U/^{206}Pb$ ratios. We acknowledge funding of our new LA–ICP–MS facility through Swiss





National Science Foundation, project 206021_170722, to Daniela Rubatto and Thomas Pettke. The solution ICP–MS isotope data was obtained on a Neptune MC–ICP mass spectrometer acquired with funds from the NCCR PlanetS supported by the Swiss National Science Foundation grant nr. 51NF40-141881. This work is part of the PhD thesis of Veronica Peverelli, who acknowledges SNF funding (project nr. 178785) granted to Alfons Berger.

**Competing interests**

The authors declare that they have no conflict of interest.

**Code/data availability**

All data are included in the manuscript (see tables).

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
