# Peer review of "U-Pb geochronology of epidote by LA-ICP-MS as a tool for dating hydrothermal-vein formation"

_Geochronology, 2020_

## Referee Comment (RC1) · Elizabeth Catlos (Referee) · 17 Oct 2020

This paper presents results of LA-ICP-MS dating of epidote veins from three geological settings: Albula (eastern Swiss Alps), Grimsel (central Swiss Alps), and along the Heyuan fault (China). The goals are to demonstrate the viability of epidote LA-ICP-MS dating and provide ages that are useful in interpreting the tectonics of these regions. The paper describes both technique development and application. In terms of technique development, the paper does not seek to reproduce epidote ages from known standards. Instead, it uses radiogenic allanite (an epidote group mineral) as a standard to date epidote of unknown ages. Overall, their use of the Terra-Wasserburg plot

appears useful and applies to epidote, which contains low amounts of radiogenic elements and higher amounts of common Pb. It is promising that the unknown LA-ICP-MS ages yields results consistent with the tectonics of these regions. However, two issues to address are: (1) the secondary standards (CAP and AVC) appear to yield ages >10 m.y. older than what is reported for these grains (Table 3), and (2) the primary standard (Tara) is heterogeneous in age (e.g., Liao et al. 2020 JAAS). Overall, the approach described here would be of use to those seeking to generate epidote ages from regions that have experienced a simple tectonic history (i.e., epidote crystallization and no subsequent deformation/recrystallization events).

Some specific comments: Lines 52-53: "However, none of these techniques are in situ, and they cannot target different microstructural and textural domains." It is possible to microdrill different domains of epidote, especially as it forms larger crystals and zones in the types of rocks under investigation. The authors also date one of the unknown epidote grains using solution ICP-MS, which seems at odds with this statement. Lines 104-107: "However, in recent years, magmatic allanite has been successfully characterized and dated by U–Th–Pb LA–ICP–MS (e.g. Gregory et al., 2007; 2012; El Korh, 2014; Smye et al., 2014), and several allanite samples have been proposed as suitable primary reference materials (e.g. Gregory et al., 2007; Smye et al., 2014)." Allanite has been shown to reproduce TIMS ages using SIMS as early as 2000 (Catlos et al., 2000, Am. Min.). Lines 110-115: "The possible issues in the use of allanite as reference material for accurate U–Th–Pb geochronology are mostly related to local isotopic heterogeneity, excess $^{206}$Pb due to incorporation of $^{230}$Th during crystallization, variable contents of initial Pb and disturbance of the geochronometer by secondary processes (e.g., hydrothermal alteration; Gregory et al., 2007; Darling et al., 2012; Smye et al., 2014; Burn et al., 2017). Nevertheless, these issues can be largely avoided by careful selection of spot analyses referring to backscattered electron (BSE) images, and by identifying and excluding problematic analyses from calculations." Allanite can be incredibly complex, as shown in numerous BSE images available throughout the literature. It is likely that a careful understanding of its chemistry in terms of compositional

analyses could be included here. I don't think any of the issues described above can be avoided, and the authors should be more precise in their description of problematic issues. Lines 128-129: "Sampling locations are respectively shown in Fig. 1 of Şengör (2016), in Fig. 1 of Wehrens et al. (2017), and in Figs. 1 and 2 of Tannock et al. (2020a)." It is unclear if the sample numbers also correlate to samples described in these references. They should perhaps provide GPS locations if the authors do not want to make geological maps of the areas. Lines 130-131: "The Albula area was chosen because the weak Alpine overprint allows for the hypothesis that epidote veins were not geochemically and isotopically altered after their formation." An overprint would suggest that alteration is possible. "Weak" is a relative term—it would be helpful to include particular P-T conditions. Lines 132-133: "The Heyuan Fault was selected because structural constraints allow to assess if the calculated U–Pb age is reasonable for epidote crystallization." Anticipated absolute ages for the samples are not provided. It would be helpful to indicate what they could be here or in further paragraphs. Lines 157-158: "As these veins are only visible within the tunnel, their relationships with Alpine structures and between each other are not understood." If the epidote ages from this area are unknown is at odds with the impression of this manuscript up until this point that the technique will reproduce epidote ages to a degree of helpful precision. Lines 201-203: "...the most homogenous allanite in terms of U–Th–Pb isotopes (Gregory et al., 2007; Burn et al., 2017) and the most promising reference material for U–Pb geochronology (Smye et al., 2014)." One would hope that all of the standards are homogenous in terms of U-Th-Pb isotopes, which translate to their ages. The CAP and AVC allanites are incredibly useful materials. They are compositionally heterogeneous, but they are incredibly well-characterized in terms of their ages. There is no need for qualitative judgments regarding allanite standards. Lines 203-205: "Tara allanite reference isotopic ratios and their uncertainties (Table 2) were calculated by averaging five ID–TIMS measurements reported by Smye et al. (2014), excluding the measurement that yielded the youngest U–Pb age outside 205 uncertainty (Smye et al., 2014)." No need to repeat the Smye reference twice in the sentence. The age and uncertainty that

they obtained from the standard should be reported. From my understanding, the Tara allanite age has a more considerable uncertainty compared to CAP and AVC (e.g., see Liao et al. 2020 JAAS). See also their concerns about the homogeneity of this grain and problems regarding the reproducibility of its ages. All of the allanite standards have ages that differ significantly from the age of the "unknowns." It would be helpful to comment on this observation. Line 302-303: "BSE images of epidote (Fig. 2a) reveal growth zoning and intra-grain veinlets resulting from interaction with a secondary fluid. Sample Albula-1 was selected for solution ICP–MS given the large size of epidote grains." This sample appears affected by secondary fluid interaction, but that was not considered problematic in terms of interpreting its solution ICP-MS result. Line 343. "...CAP$^b$." I am not sure what CAP superscript b is exactly. It is likely explained somewhere, but I could not find it. Line 445: "overall consistent with published U–Pb ages, attesting to reliable U–Pb measurements..." It would be helpful to indicate what the U-Pb ages are and reference those results. Lines 512-13: "These considerations confirm the Tera–Wasserburg approach as the most suitable – and often the only viable – for accurate U–Pb dating of low-$\mu$ phases such as epidote (see Romer, 2001; Romer and Xiao, 2005)." The approach applied is very similar to that done for apatite by others (Oldum and Stockli, 2019, Tectonics and 2020 EPSL). Table 1. Is difficult to read. Similar parameters for each session can be placed in a footnote. There is a footnote b at the last row that is not explained. Table 2. Reference data for the other standards could be provided as well. Please include the ages. Table 3. The CAP and AVC ages reported here are 10 m.y. older than their TIMS ages (Barth, 1994, also Liao et al. 2020). Table 4 and Table 5. Are difficult to read in terms of which dataset belongs to which sample. Could place the sample names in rows. Figure 2. Are the BSE images located in the petrographic images in Figure 1? No spots are shown in panel a1, so I assume it was not also dated using LA-ICP-MS. I think the contrast can be enhanced in some of the images to enhance the zoning.

---

## Referee Comment (RC2) · James Darling (Referee) · 16 Nov 2020

General comments

The manuscript presents new data from in-situ U-Pb isotopic analysis of epidote in a series of hydrothermal veins, using modified LA-ICP-MS protocols. The method and results show great untapped potential for epidote (mineral) geochronology, building upon previous studies of allanite (REE-rich epidote supergroup mineral). Given the occurrence of epidote in a variety of important geological settings, the developments have great potential to improve the geochronology of crustal processes.

[Figure]

The manuscript documents an approach to correcting laser induced elemental fractionation using an allanite primary reference material. This shows excellent promise, and the resulting ages determined for epidotes in vein samples seem geologically reasonable. The manuscript is generally well-written and supported by some useful figures and tables (although there is room for improvement). Overall, I consider that the manuscript will be very well suited to Geochronology, following some modifications.

I have made a number of specific comments below that are presented in order of reading through the text. These can be broadly grouped into three areas that need some consideration:

(1) The documentation of downhole fractionation (DF) correction and error propagation needs to be clarified and expanded upon. Please see comments below that highlight inconsistencies between the text and Figures relating to DF corrected data, and which require a more balanced discussion of results.

(2) There are subtleties in the Grimsel-1 data that have not been discussed in the text, and may have significant implications for the interpretation of epidote ages in veins with complex histories. Please see comments listed below.

(3) Greater clarity is needed on the distinction on epidote (mineral) versus epidote group/supergroup minerals. This is particularly highlighted by the overlap and comparison with studies of allanite (REE-rich monoclinic epidote). The novely here is specifically in measurement of epidote (mineral), although solid-solution with clinozoisite should be acknowledged. This is particularly important given that chemical data (e.g. Al, REE, Fe) to demonstrate near end-member compositions has not been measured for the studied grains. Please see suggestions listed below.

Specific comments:

Lines 44-51: The text needs to make a clearer distinction between epidote supergroup minerals and epidote here. Oberli et al., (2004) measured grains varying from low to v.

high REE+Th contents (i.e. "allanite")

Lines 45-46: epidote supergroup minerals can have wt. % levels of Th (i.e. allanite)

Lines 58-60: several papers have presented methods for monoclinic epidote group minerals (typically those with a high allanite component). Given that the supergroup contains sold-solution series, the distinction between epidote and epidote supergroup minerals needs to be made much clearer here - especially as magmatic allanite used as primary reference material - doesn't really seem right to say that no one has done this before unless you more specifically mean end-member epidote. Could also mention here that the protocols in this study are very similar to those applied to apatite.

See also Conclusion Line 564 and elsewhere in the text, where use of 'monoclinic epidote' is a bit vague in this regard (as this could include allanite).

Lines 99-103: It has also been shown that DF can be minimized to the point of not requiring matrix-matched standardization (https://doi.org/10.1016/j.chemgeo.2011.11.012). It would be good to acknowledge this here, because it provides an alternative approach for U-Th-Pb isotope analysis of allanite (epidote supergroup minerals).

Lines 107-110: again, here the text gets a bit muddled between epidote supergroup minerals and epidote group minerals (for the latter it is claimed that no previous geochron work has been undertaken in Line 58). Epidote supergroup minerals with a high allanite component are also monoclinic - so use of 'monoclinic epidote' here is not clear in meaning.

Lines 115-118: it is incorrect to say that 204Pb corrections are not possible from LA-ICP-MS data. See https://doi.org/10.1016/j.chemgeo.2011.11.012 and especially Cenki-Tok et al., (2014: https://doi.org/10.1111/ter.12066).

Lines 117-121: I think that a statement should be added here making it clear that it has been shown in many papers that Stacey & Kramers model values are often not

appropriate for correcting allanite, titanite, rutile etc U-Pb ratios, and hence extreme caution is required here.

Lines 215-220: detail of the DF corrections applied is missing here. How was DF modelled and corrected in Iolite?

Lines 224-226: Do the final uncertainties provided for unknowns include propagated uncertainties from the 207Pb-correction of Tara (including uncertainty in initial Pb composition and correction)? Please specify. If not, these sources of uncertainty should be fully propagated through to the results.

Lines 229-234: Please summarize here or in Table 3 the effect on precision of anchoring the 207Pb/206Pbc.

Section 4.1: No mineral chemistry is presented for the studied epidotes. Are there independent constraints on the composition of these grains? Given solid solution with clinozoistite, it would be useful to know if there is any relationship between major element chemistry and U/Th/Pb contents, as well as possible links with matrix effects.

Lines 324-345: As above, how was DF corrected in iolite? I've not seen details documented anywhere as yet (function used?).

Section 4.2 & Figure 4: The discussion of DF in unknowns is very cursory and some additional analysis seems to be warranted by the data. The text states that all of the unknowns have 'parallel flat lines' on Figure 4, but this is not correct. Focusing on Figure 4C, there are analyses that have decreasing ratios through time, and others that have increasing ratios. This indicates that the assumption of exact matrix matching between Tara and all of the unknowns is not perfect. To me, it seems likely that the DF correction is working within the large uncertainties of individual measurements, but a more detailed analysis of this issue is warranted. What are the differences between analyses with +ve and -ve slopes here (compositional?) and what is the likely effect on accuracy and uncertainties? These issues need to be acknowledged in the main

[Figure]

text and at a minimum state that the DF correction seems to be working within large uncertainties of individual epidote measurements. - it is possible that this variability is caused by zonation in concentration, rather than a matrix effect. The presentation of data on Figure 4 is not very clear, which limits the ability to really resolve these issues. You could present these as % change in the ratio through time, and either select a subset of analyses that have independent measures of heterogeneity or average data from each X second time interval since shutter opening.

Figure 5: as per previous comment, some of the ablations shown are certainly not "flat" when it comes to DF-corrected 206Pb/238U. This should be acknowledged in the text and a more detailed analysis provided. At lease one measurement on Fig. 5a has huge variation in the ratio - linked to variable U +/- Pb contents, or weird ablation behaviour?

Data for Grimsel 1, lines 388-398: There is a bit of an issue with the Grimsel-1 data here. For the 30 micron data, the text states that 4 data points were rejected on the grounds that they 'cause higher MSWDs'. However, it is correctly noted in the intro to T-W plots that scatter can reflect non-cogenetic origins (and hence have important geological meaning). To test this, I replotted the 30 micron data using IsoplotR; using all 25 data points I get a resulting T-W intercept age of 17.25 +/- 11.15 Ma (95 % conf.; MSWD = 1.2). I do not see any obvious reason to exclude any of this data (especially as IsoplotR includes scatter in the 95% conf. uncertainties). One issue, the 7/6 intercept on my plot is 0.7863 +/- 0.0051, which is JUST outside of uncertainty of the initial on Figure 6b. Could this reflect either (a) underestimation of uncertainty in the 50 micron data (note MSWD <1) or sampling of external Pb using larger spot sizes (i.e. modern lab Pb)? Please replot the 30 micron data to check all of this, and I don't think that grounds to exclude points are strong.

Following on from that, why not combine the 50 and 30 micron data for Grimsel-1 into a combined T-W? I did this, and get a result of 15.69 +/- 5.94 Ma (95 % conf.; MSWD = 2.6), with initial 7/6 of 0.7922 +/- 0.0033. The distinction could be important, as the ages for ductile deformation in the area from Rolland et al., (2009) are ∼21 Ma (Stage

1) and ∼14-12 Ma (stage 2), and these authors speculate that brittle structures formed at ∼ 15.5 Ma. Could the higher MSWD of this regression be reflecting some epidote growth/resetting throughout this complex deformation history?

Lines 466-467: change 'used to normalize the measured isotopic ratios to real values after correcting them for DF' to used to correct measured isotopic rations for DF.

Lines 467-470: as per previous comment, the corrected ratios shown are not all "flat", so this needs to be changed and a more complete analysis of DF corrected ratios presented.

Section 5.3: I found this section quite repetitive, and some of the key points (spread & sample volume) have already been made in the previous section. It would be useful to restructure and refine Sections 5.2 and 5.3 to produce a more focused and less repetitive discussion.

Section 5.4: The Cenki-Tok et al., (2014) paper provides an excellent example of the need to independently determine initial Pb compositions to correct allanite analyses. I reccomend mentioning that study at this point of the discussion.

Lines 523-529: as per previous comment, the existing geochron in the Grimsel area is a bit more complex than shown in the discussion here. Rolland et al., (2009) document two distinct ductile deformation phases at ∼21 Ma and 14-12 Ma - is there particular evidence to suggest that the epidote bearing veins are only recording the earlier episode? Perhaps epidotes in these folded veins are being partially reset during the younger ductile event?

Lines 545-555: The discussion of the Grimsel vein results may need tweaking given the slightly younger age determined from the combined 50 and 30 micron spot data. Unless there is a clear reason not to combine these datasets, the slightly younger age and higher MSWD could have bery interesting implications for the significance of epidote ages from these samples....

Other technical suggestions:

Abstract, line 8: should read 238U (rather than 283U)

Abstract, lines 14-15: Would be useful for the text here to be a bit more specific on what is meant by 'appreciable', and also which aspects of the initial Pb are variable (presumably this primarily refers to concentration?

Abstract, Lines 20-21: It is possible for epidotes in a sample to be cogenetic (formed during the same event) and still record variable initial Pb isotope compositions, e.g. https://doi.org/10.1007/s00410-003-0494-6

Lines 80-84: Need to split this into two sentences.

Line 93: minerals

Lines 83-96: there is a lenghty description here of the Tera-Wasserburg diagram approach. Given that this is widely used in the accessory mineral geochron community, perhaps this description is not all needed and instead the text could focus on issues relating to epidote geochron more specifically (e.g. U contents, initial Pb variability) . There also is some repetition here (fraction of initial Pb; upper 207Pb/206Pb).

Figure 6 caption: there are two (b)s and no (c) listed

Figure 7 caption: what did the MatLab script do?

Lines 129-133: Given that detail of these regions comes in subsequent paragraphs, I'd reccomend changing this to a broader statement of motivation - i.e. targeted regions with well-constrained histories. Some more specific issues are teased here (e.g. alteration), but without key citations.

Table 1: for which material are the sensitivity figures provided? These would be better provided as cps/ppm (if a homogenous material).

---

## Author Response (AR2)

Dear Editor, dear Associate Editor,

All corrections required by the associate editor on 20 January 2021 have been made to the revised manuscript. Below is the point-to-point response submitted on 17 January 2021.

Thank you for having considered our manuscript "U–Pb geochronology of epidote by LA–ICP–MS as a tool for dating hydrothermal-vein formation" for publication.

Yours sincerely,
Veronica Peverelli, on behalf of all authors

Dear Editor,

Please find below a point-to-point response to all comments by the referees and the associate editor following the Associate Editor Decision on our manuscript "U–Pb geochronology of epidote by LA–ICP–MS as a tool for dating hydrothermal-vein formation" (MS id: gchron-2020-27).

This point-to-point response addresses the associate editor comments first, giving a response in RED to indicate how the text was revised. The referee comments follow, laid out as in the Author's responses (submitted on Dec. 9th, 2020). Short additional responses/notes are added in RED to confirm the changes.

The following color code applies:
GREEN = General issue; BLUE = Referee/associate editor comment; BLACK = Response to comments by referees (before Associate Editor Decision); RED = Response/note to comments by referees and associate editor (after Associate Editor Decision).

All suggestions made by the referees and by the associate editor have been taken into account during the revision. The manuscript has been proofread and rephrased for clarity where necessary.

We hope that you will find the revised manuscript satisfactory.

Yours sincerely,
Veronica Peverelli, on behalf of all authors

**Response to comments by Axel Schmitt (Associate Editor)**

1. more clarity regarding the reference materials (as argued for by reviewer 1) and the treatment of downhole fractionation (as raised by reviewer 2) will require careful attention.
   Done.

2. Line 80: "If the isotopic composition of initial Pb is unknown, 204Pb contents cannot be precisely determined..." The phrasing of this sentence is confusing (it can be read as if knowing initial – or common – Pb is a prerequisite to precisely measuring 204Pb). I therefore suggest: "This, however, is not always possible, either because the isotopic composition of initial Pb is unknown or 204Pb contents cannot be precisely determined (e.g. because of the analytical technique employed). Hence, if no other dating method is viable (e.g. too-low Th contents hampering Th–Pb dating), the best solution for U–Pb dating of low-μ phases is using a regression in a Tera–Wasserburg diagram (Tera and Wasserburg, 1972) that plots measured 207Pb/206Pb versus 238U/206Pb ratios.
   The sentence was rephrased.

3. Line 93: minerals
   Done.

4. Line 97: second "may" not needed
   Done.

5. Line 272: Where does n = 15 come from? This would need a citation of a justification.
   The sentence was rephrased.

6. Line 276: Omit the first "as possible"
   Done.

7. Line 297: wrap = mantle (?)
   Done.

8. Fig. 4: Legend needs to be improved. This should mention what individual lines are.
   The following was added to the caption text: "Each individual line represents one analysis".

9. Line 372: present = stated
   Done.

10. Fig. 6: Error envelopes should also be plotted here (see Fig. 7)
    Done.

11. Line 389/397: poor/poorly appears to be judgmental; more neutral/descriptive wording would be appropriate. Use: "nearly invariable" or words like "limited" or "minor".
Done.

12. Line 424: Same here with "meager"
Done.

13. Fig. 7: In agreement with reviewer 2, I wonder how the error envelope was calculated, and why the commonly used scheme of Ludwig (1980) was not used? (Ludwig, K. R., 1980, Calculation of uncertainties of U-Pb isotope data. Earth and Planetary Science Letters, 46-2, 212-220).
All error envelopes were (re-)plotted using Isoplot.

14. Line 493: high-enough = sufficient
Done.

**Author's response to referee comments by Elizabeth Catlos (Referee #1)**

General issues:

1. Comments on Tara allanite as primary reference material
We acknowledge the complications of allanite as a standard. Nevertheless, all allanite samples used in this work have been characterized in the cited literature. It is the conclusion of the cited studies that Tara allanite is the most isotopically homogenous among our available allanite samples and therefore the most promising reference material for U-Th-Pb dating. Moreover, regarding Tara allanite Liao et al. (2020; section 5.1) state that "the relatively low common Pb concentration ($f_{206} \sim$ 7–20% and $f_{208} < 1\%$) and good U/Th–Pb age reproducibility demonstrate its applicability as a U–Th–Pb dating standard".
Smye et al. (2014) report the only available ID-TIMS data for Tara allanite. The fact that its reference age is a matter of debate can indeed be problematic, as pointed out by the referee in "Comment #7". However, the fact that our allanite secondary standards yield accurate U-Pb ages indicates that the reference values from Smye et al. (2014) are accurate for Tara allanite. We therefore disagree with the referee's concerns on the suitability of Tara allanite as primary reference material.
Added Liao et al. (2020) as reference.

Referee comments by Elizabeth Catlos:

1. "It is possible to microdrill different domains of epidote, especially as it forms larger crystals and zones in the types of rocks under investigation. The authors also date one of the unknown epidote grains using solution ICP-MS, which seems at odds with this statement." (lines 52-53)
The epidote micro-separate analyzed by solution ICP-MS was in fact not dated. It was only used to inspect the consistency between LA- and solution ICP-MS data in a Tera-Wasserburg diagram. The revised text will however be edited acknowledging the possibility of micro-drilling material for *in-situ* TIMS and Pb-Pb dating.
Done.

2. "Allanite has been shown to reproduce TIMS ages using SIMS as early as 2000 (Catlos et al., 2000, Am. Min.)" (lines 104-107)
The suggested reference will be added.
Done and rephrased text.

3. "Allanite can be incredibly complex, as shown in numerous BSE images available throughout the literature. It is likely that a careful understanding of its chemistry in terms of compositional analyses could be included here. I don't think any of the issues described above can be avoided, and the authors should be more precise in their description of problematic issues." (lines 110-115)
The compositional characterization of the used allanite samples is presented in the cited literature. The statement that none of the issues related to allanite for its use in geochronology can be avoided is at odds with what concluded in the cited literature, and we therefore disagree with the referee's comment. It is in fact beyond the scope of this study to address the issues related to allanite geochronology, but rather to prove that it is a suitable reference material for epidote U-Pb geochronology.

4. "It is unclear if the sample numbers also correlate to samples described in these references. They should perhaps provide GPS locations if the authors do not want to make geological maps of the areas." (lines 128-129)
Samples Albula-1, Grimsel-1 and Grimsel-2 presented in this study were not characterized elsewhere, and the cited literature and figures are only intended to show the sampling locations. Sample Heyuan-1 is instead discussed in Tannock et al. (2020a; 2020b; their sample HY17-5), with the exact sampling location shown in Fig. 1 of Tannock et al. (2020a). This will be clarified in the revised text. In addition to this, GPS locations of sampling localities will be provided or it will be specified where they are reported in the cited literature.
Done.

5. "An overprint would suggest that alteration is possible. "Weak" is a relative term. It would be helpful to include particular P-T conditions." (lines 130-131)
Specific conditions will be added to the revised text.
Done.

6. "Anticipated absolute ages for the samples are not provided. It would be helpful to indicate what they could be here or in further paragraphs." (lines 132-133)
"If the epidote ages from this area are unknown is at odds with the impression of this manuscript up until this point that the technique will reproduce epidote ages to a degree of helpful precision." (lines 157-158)
The revised text will be rephrased in a way to make it clear that there are no precise anticipated ages for the epidote samples, but that the consistency of the obtained ages can be verified thanks to the well-known tectonic history of the sampling areas.
Done.

7. "One would hope that all of the standards are homogenous in terms of U-Th-Pb isotopes, which translate to their ages. The CAP and AVC allanites are incredibly useful materials. They are compositionally heterogeneous, but they are incredibly well-characterized in terms of their ages. There is no need for qualitative judgments regarding allanite standards." (lines 201-203)
See "General issue #1".

8. "No need to repeat the Smye reference twice in the sentence. The age and uncertainty that they obtained from the standard should be reported. [continues below]
The repeated reference will be deleted, and age and uncertainty of Tara allanite obtained by Smye et al. (2014) will be added to Table 2.
Done.
From my understanding, the Tara allanite age has a more considerable uncertainty compared to CAP and AVC (e.g., see Liao et al. 2020 JAAS). See also their concerns about the homogeneity of this grain and problems regarding the reproducibility of its ages. [continues below]
See "General issue #1".
All of the allanite standards have ages that differ significantly from the age of the "unknowns." It would be helpful to comment on this observation." (lines 203-205)
See "Comment #15".

9. "This sample appears affected by secondary fluid interaction, but that was not considered problematic in terms of interpreting its solution ICP-MS result." (lines 302-303)
A few laser spots were intentionally placed on areas with veinlets, as well as across zonations, to assess whether these features would yield outliers within the dataset. We conclude that they do not based on the statistical values obtained from the Tera-Wasserburg diagrams, and

consequently that sample Albula-1 could be safely analyzed by solution ICP-MS. This will be discussed in sections 3.2 and 4.3 of the revised manuscript.
Done.

10. I am not sure what CAP superscript b is exactly. It is likely explained somewhere, but I could not find it. (line 343)
Burn et al. (2017) sampled CAP[b] allanite from the Cima d'Asta pluton and determined ages consistent with those presented for CAP allanite by Barth et al. (1994). They therefore concluded that it is likely that CAP[b] and CAP are the same allanite, but could not confirm this from field relations – hence the superscript in CAP[b]. This will be made clearer in the revised manuscript.
It has been indicated in the text that the details can be found in Burn et al. (2017).

11. "It would be helpful to indicate what the U-Pb ages are and reference those results." (line 445)
The U-Pb ages will be mentioned in the revised text.
Done.

12. "The approach applied is very similar to that done for apatite by others (Oldum and Stockli, 2019, Tectonics and 2020 EPSL)" (lines 512-513)
The suggested literature will be cited.
Done.

13. "Table 1. Is difficult to read. Similar parameters for each session can be placed in a footnote. There is a footnote b at the last row that is not explained."
The table will be revised to improve clarity. A typo will be corrected (i.e. ablation time with 30-micron spot size is 30 sec and not 40 sec). See "Comment #10" for the footnote.
Done.

14. "Table 2. Reference data for the other standards could be provided as well. Please include the ages."
The table will be edited accordingly.
The ages were added to Table 2.

15. "Table 3. The CAP and AVC ages reported here are 10 m.y. older than their TIMS ages (Barth, 1994, also Liao et al. 2020)."
TIMS ages are actually Th-Pb ages. It is in fact reported and thoroughly addressed in the cited literature that U-Pb ages of allanite tend to be older than their reference Th-Pb ages, and addressing this issue is outside the scope of this work. Since we only measure U-Pb ages, we compare U-Pb ages from allanite secondary reference materials to the available literature in order to check for reproducibility of U-Pb ages, rather than for their concordance with Th-Pb

ages. Indeed, one CAP U-Pb age that we obtained (July 2019 session) is slightly outside uncertainty of its U-Pb age reported by Gregory et al. (2007). Although we recognize that this specific age alone may raise concerns, the other secondary reference material (AVC allanite) used in the same session gave a U-Pb age consistent with its U-Pb age reported by Gregory et al. (2007). A brief statement of these points will be added to the manuscript.
Done.

16. "Table 4 and Table 5. Are difficult to read in terms of which dataset belongs to which sample. Could place the sample names in rows."
The tables will be improved in clarity in the revised manuscript.
Done.

17. "Figure 2. Are the BSE images located in the petrographic images in Figure 1? No spots are shown in panel a1, so I assume it was not also dated using LA-ICP-MS. I think the contrast can be enhanced in some of the images to enhance the zoning."
We do not show the exact location of the BSE image within the petrographic ones, but will consider adding rectangles in the latter. The BSE image of panel 1 shows an epidote grain where all described features (growth zoning, veinlets and fractures) can be seen, but this specific grain was not analyzed. Contrast and brightness were optimized upon image acquisition: the zoning in epidote grains is simply modest, and thus faint in the BSE images.
Rectangles were added in Fig. 1 and the caption of Fig. 2 was made clearer.

**Author's response to referee comments by James Darling (Referee #2)**

General issues:

1. Documentation of correction for downhole fractionation
The referee points out that in Fig. 4b,c and 5 some of the lines representing time-resolved (total) $^{206}$Pb/$^{238}$U ratios are not flat despite having been corrected for downhole fractionation. This is indeed true, and after re-evaluation of the data we have established that this is due to zoning in initial Pb contents in most cases. This was verified by applying a $^{208}$Pb correction with values calculated from the Tera-Wasserburg diagram. After this, most of the sloped lines become flat, demonstrating that the correction for downhole fractionation worked properly. However, some analyses still displayed some sloping in the time-resolved (radiogenic) $^{206}$Pb/$^{238}$U ratios even after the application of the $^{208}$Pb correction. We therefore decided to exclude these suspicious analyses from the dataset for age calculation, and obtained ages and initial $^{207}$Pb/$^{206}$Pb ratios that are within uncertainty of those presented in the first version of the manuscript. The effects of variable initial Pb contents on the accuracy of analyses as well as the need to carefully assess $^{208}$Pb-corrected time-resolved $^{206}$Pb/$^{238}$U ratios after correction for

downhole fractionation will be discussed in section 4.2 of the revised version. Updated figures, ages and isotopic ratios will be included in the revised manuscript.

Done. Figs. 4 and 5 were updated (Fig. 4 only showing secondary reference materials and Fig. 5 only showing epidote unknowns). An explanation of the above is given in the text.

2. Sample Grimsel-1 data

Following "General issue #2", some analyses were excluded from the dataset for age calculation. We appreciate the suggestion of the referee to combine the 50- and 30-micron datasets, and this possibility will be addressed in the revised manuscript. We also considered the proposed geological interpretation of the obtained ages as reflecting partial reset of this epidote sample during subsequent deformation. We recognize that sample Grimsel-1 is texturally and geochronologically complex. A work on epidote microstructures and their relationships with trace element and isotopic data is currently being carried out, and it would be outside the scope of this methodological contribution. However, we agree with the referee that partial resetting cannot be ruled out, and we will hence discuss the referee's hypothesis in the revised text.

The interpretation of sample Grimsel-1 was updated and the referee's comment was taken into account (see Sects. 4.3, 5.5. and 5.6).

3. Comments on the use of the phrase "monoclinic epidote"

Our epidote samples are compositionally within the epidote-clinozoisite solid solution and we agree that the phrase "monoclinic epidote" is inaccurate. A clearer definition of the use of the term "epidote" will be given in the revised text and the manuscript will be edited accordingly. The fractions of epidote ($X_{Epi}$) and clinosoizite ($X_{Czo}$) components will be added in section 4.1.

$X_{Epi}$ and $\Sigma REE$ were given in Sect. 4.1. $X_{Epi}$ was measured by EPMA, and therefore details of the setup were added to Appendix A.

Referee comments by James Darling:

1. "The text needs to make a clearer distinction between epidote supergroup minerals and epidote here. Oberli et al., (2004) measured grains varying from low to high REE+Th contents (i.e. "allanite")" (lines 44-51)

See "General issue #3". Oberli et al. (2004) also report one analysis on epidote (i.e. epidote-clinozoisite solid solution; their section 5.3 and their figure 5). It will be made clear in the revised text that we refer to this one analysis.

Done.

2. "epidote supergroup minerals can have wt. % levels of Th (i.e. allanite)" (lines 45-46)

See "General issue #3".

3. "several papers have presented methods for monoclinic epidote group minerals (typically those with a high allanite component). Given that the supergroup contains sold-solution series, the distinction between epidote and epidote supergroup minerals needs to be made much clearer here - especially as magmatic allanite used as primary reference material - doesn't really seem right to say that no one has done this before unless you more specifically mean end-member epidote. Could also mention here that the protocols in this study are very similar to those applied to apatite. See also Conclusion Line 564 and elsewhere in the text, where use of 'monoclinic epidote' is a bit vague in this regard (as this could include allanite)." (lines 58-60)
   See "General issue #3". The similarity with the protocols applied to apatite will be acknowledged in the revised text.
   Done.

4. "It has also been shown that DF can be minimized to the point of not requiring matrix-matched standardization (https://doi.org/10.1016/j.chemgeo.2011.11.012). It would be good to acknowledge this here, because it provides an alternative approach for U-Th-Pb isotope analysis of allanite (epidote supergroup minerals)." (lines 99-103)
   The possibility of non-matrix-matched standardization will be mentioned in the revised text.
   Done.

5. "again, here the text gets a bit muddled between epidote supergroup minerals and epidote group minerals (for the latter it is claimed that no previous geochron work has been undertaken in Line 58). Epidote supergroup minerals with a high allanite component are also monoclinic - so use of 'monoclinic epidote' here is not clear in meaning." (lines 107-110)
   See "General issue #3".

6. "it is incorrect to say that 204Pb corrections are not possible from LA-ICP-MS data. See https://doi.org/10.1016/j.chemgeo.2011.11.012 and especially Cenki-Tok et al., (2014: https://doi.org/10.1111/ter.12066)." (lines 115-118)
   Agreed. We will rephrase the text to clarify that a $^{204}$Pb correction is complex and we thus prefer to use the Tera-Wasserburg approach, which uses ratios that are not corrected for initial Pb. We will also acknowledge the possibility of correcting for initial Pb by measuring $^{204}$Pb in coexisting mineral as done by Cenki-Tok et al. (2014).
   Done.

7. "I think that a statement should be added here making it clear that it has been shown in many papers that Stacey & Kramers model values are often not appropriate for correcting allanite, titanite, rutile etc U-Pb ratios, and hence extreme caution is required here." (lines 117-121)
   We agree. In fact, we do not correct Tara allanite by using Stacey & Kramers (1975) model values – as erroneously stated in the text. Instead, we determined the initial $^{207}$Pb/$^{206}$Pb ratio from a Tera-Wasserburg diagram that we created based on the TIMS data reported by Smye et

al. (2014). This was the value used to apply the $^{207}$Pb correction to Tara allanite. This value is already reported in Table 2, and the text will be edited accordingly.
Done.

8. "detail of the DF corrections applied is missing here. How was DF modelled and corrected in Iolite?" (lines 215-220)
We selected an exponential function to correct for downhole fractionation in Iolite. Iolite fits this functions to model the downhole fractionation on primary reference materials, then applies this function to all unknown analyses to correct them for downhole fractionation. This will be specified in the revised text.
Done.

9. "Do the final uncertainties provided for unknowns include propagated uncertainties from the 207Pb-correction of Tara (including uncertainty in initial Pb composition and correction)? Please specify. If not, these sources of uncertainty should be fully propagated through to the results." (lines 224-226)
The VisualAge_UcomPbine package used on Iolite for data reduction is described by Chew et al. (2014). Although the authors do not specify whether or not the uncertainty coming from initial Pb correction is propagated onto the unknowns, an overall uncertainty coming from the reproducibility of the primary reference material is propagated by Iolite.
Specified in the revised text.

10. Please summarize here or in Table 3 the effect on precision of anchoring the 207Pb/206Pbc. (lines 229-234).
This will be done in the revised manuscript.
Unanchored regression ages are given in Table 3.

11. "No mineral chemistry is presented for the studied epidotes. Are there independent constraints on the composition of these grains? Given solid solution with clinozoistite, it would be useful to know if there is any relationship between major element chemistry and U/Th/Pb contents, as well as possible links with matrix effects." (section 4.1)
A study on mineral chemistry is part of ongoing work. A consideration of matrix effects related to mineral chemistry, although captivating, goes beyond the scope of this contribution. From the present data it can nonetheless be stated that, since MSWD is low in all our Tera-Wasserburg diagrams, matrix effects – if any – have no effect at the current analytical precision.

12. "As above, how was DF corrected in iolite? I've not seen details documented anywhere as yet (function used?)." (lines 324-345)
See "Comment #8".

13. "The discussion of DF in unknowns is very cursory and some additional analysis seems to be warranted by the data. The text states that all of the unknowns have 'parallel flat lines' on Figure 4, but this is not correct. Focusing on Figure 4C, there are analyses that have decreasing ratios through time, and others that have increasing ratios. This indicates that the assumption of exact matrix matching between Tara and all of the unknowns is not perfect. To me, it seems likely that the DF correction is working within the large uncertainties of individual measurements, but a more detailed analysis of this issue is warranted. What are the differences between analyses with +ve and -ve slopes here (compositional?) and what is the likely effect on accuracy and uncertainties? These issues need to be acknowledged in the main text and at a minimum state that the DF correction seems to be working within large uncertainties of individual epidote measurements. - it is possible that this variability is caused by zonation in concentration, rather than a matrix effect. The presentation of data on Figure 4 is not very clear, which limits the ability to really resolve these issues. You could present these as % change in the ratio through time, and either select a subset of analyses that have independent measures of heterogeneity or average data from each X second time interval since shutter opening." (section 4.2 and figure 4)

"as per previous comment, some of the ablations shown are certainly not "flat" when it comes to DF-corrected 206Pb/238U. This should be acknowledged in the text and a more detailed analysis provided. At lease one measurement on Fig. 5a has huge variation in the ratio - linked to variable U +/- Pb contents, or weird ablation behaviour?" (figure 5)

See "General issue #1".

14. "There is a bit of an issue with the Grimsel-1 data here. For the 30 micron data, the text states that 4 data points were rejected on the grounds that they 'cause higher MSWDs'. However, it is correctly noted in the intro to T-W plots that scatter can reflect non-cogenetic origins (and hence have important geological meaning). To test this, I replotted the 30 micron data using IsoplotR; using all 25 data points I get a resulting T-W intercept age of 17.25 +/- 11.15 Ma (95 % conf.; MSWD = 1.2). I do not see any obvious reason to exclude any of this data (especially as IsoplotR includes scatter in the 95% conf. uncertainties). One issue, the 7/6 intercept on my plot is 0.7863 +/- 0.0051, which is JUST outside of uncertainty of the initial on Figure 6b. Could this reflect either (a) underestimation of uncertainty in the 50 micron data (note MSWD <1) or sampling of external Pb using larger spot sizes (i.e. modern lab Pb)? Please replot the 30 micron data to check all of this, and I don't think that grounds to exclude points are strong. Following on from that, why not combine the 50 and 30 micron data for Grimsel-1 into a combined T-W? I did this, and get a result of 15.69 +/- 5.94 Ma (95 % conf.; MSWD = 2.6), with initial 7/6 of 0.7922 +/- 0.0033. The distinction could be important, as the ages for ductile deformation in the area from Rolland et al., (2009) are ~21 Ma (Stage 1) and ~14-12 Ma (stage 2), and these authors speculate that brittle structures formed at ~15.5 Ma. Could the higher

MSWD of this regression be reflecting some epidote growth/resetting throughout this complex deformation history?" (Data for Grimsel 1, lines 388-398)
See "General issue #2".

15. "change 'used to normalize the measured isotopic ratios to real values after correcting them for DF' to used to correct measured isotopic rations for DF." (lines 466-467)
This will be done in the revised text.
Done.

16. "as per previous comment, the corrected ratios shown are not all "flat", so this needs to be changed and a more complete analysis of DF corrected ratios presented." (lines 467-470)
See "General issue #1".

17. "I found this section quite repetitive, and some of the key points (spread & sample volume) have already been made in the previous section. It would be useful to restructure and refine Sections 5.2 and 5.3 to produce a more focused and less repetitive discussion." (section 5.3)
We will rephrase and shorten section 5.3 in the revised text. However, sections 5.2 and 5.3 address two separate issues: one dealing with the appropriateness of the correction for downhole fractionation with different spot sizes, and the other dealing with the effects of analytical uncertainty on the quality of the regression. We therefore prefer to keep them separate.
Done.

18. "The Cenki-Tok et al., (2014) paper provides an excellent example of the need to independently determine initial Pb compositions to correct allanite analyses. I reccomend mentioning that study at this point of the discussion." (section 5.4)
See "Comment #7".

19. "as per previous comment, the existing geochron in the Grimsel area is a bit more complex than shown in the discussion here. Rolland et al., (2009) document two distinct ductile deformation phases at ~21 Ma and 14-12 Ma - is there particular evidence to suggest that the epidote bearing veins are only recording the earlier episode? Perhaps epidotes in these folded veins are being partially reset during the younger ductile event?" (lines 523-529)
"The discussion of the Grimsel vein results may need tweaking given the slightly younger age determined from the combined 50 and 30 micron spot data. Unless there is a clear reason not to combine these datasets, the slightly younger age and higher MSWD could have bery interesting implications for the significance of epidote ages from these samples...." (lines 545-555)
See "General issue #2".

20. "should read 238U (rather than 283U)" (abstract, line 8)

It will be corrected in the revised text.

Done.

21. "Would be useful for the text here to be a bit more specific on what is meant by 'appreciable', and also which aspects of the initial Pb are variable (presumably this primarily refers to concentration?)" (abstract, lines 14-15)

This does refer to the contents of initial Pb relative to total Pb. The revised text will be rephrased and fractions of $^{206}$Pb added to the data.

Done.

22. "It is possible for epidotes in a sample to be cogenetic (formed during the same event) and still record variable initial Pb isotope compositions, e.g. https://doi.org/10.1007/s00410-003-0494-6" (abstract, lines 20-21)

This aspect will be acknowledged in the introduction in the revised manuscript.

Done.

23. "Need to split this into two sentences." (lines 80-84)

"minerals" (line 93)

These corrections will be made in the revised text.

Done.

24. "is a lenghty description here of the Tera-Wasserburg diagram approach. Given that this is widely used in the accessory mineral geochron community, perhaps this description is not all needed and instead the text could focus on issues relating to epidote geochron more specifically (e.g. U contents, initial Pb variability). There also is some repetition here (fraction of initial Pb; upper 207Pb/206Pb)." (lines 83-96)

The manuscript is intended for non-geochronologists as well. Therefore, some introductory information about the Tera-Wasserburg diagram should be given. We preferred to address the effects of U contents and initial Pb variability in sections 5.3 and 5.4 because the discussion can be directly related to the presented data. We will revise the text to avoid any unnecessary repetitions.

Done.

25. "there are two (b)s and no (c) listed" (figure 6, caption)

This will be corrected in the revised manuscript.

Done.

26. "what did the MatLab script do?" (figure 7, caption)

We applied a simple fit to the data using a first-order polynomial. The uncertainty envelope is at 1σ level. This will be added to the revised caption.

The error envelope was re-plotted using Isoplot.

27. "Given that detail of these regions comes in subsequent paragraphs, I'd reccomend changing this to a broader statement of motivation - i.e. targeted regions with well-constrained histories. Some more specific issues are teased here (e.g. alteration), but without key citations." (lines 129-133)

Agreed, these lines will be edited in the revised manuscript.

Done.

28. "for which material are the sensitivity figures provided? These would be better provided as cps/ppm (if a homogenous material)." (table 1)

Sensitivity was measured on NIST SRM612. cps/ppm values will be given in the revised table.

Done. Since sensitivity was measured by scanning ablation, the scan rate was also added.